# Changes in the high latitude Southern Hemisphere through the Eocene-Oligocene Transition: a model-data comparison

Alan T. Kennedy-Asser[1,2], Daniel J. Lunt[1,2], Paul J. Valdes[1,2], Jean-Baptiste Ladant[3], Joost Frieling[4], Vittoria Lauretano[2,5]

[1] BRIDGE, School of Geographical Sciences, University of Bristol, Bristol, UK
[2] Cabot Institute for the Environment, University of Bristol, Bristol, UK
[3] Department of Earth and Environmental Sciences, University of Michigan, Ann Arbor, USA
[4] Marine Palynology and Paleoceanography, Laboratory of Palaeobotany and Palynology, Department of Earth Sciences, Faculty of Geosciences, Utrecht University, Princetonlaan 8a, 3584CB Utrecht, the Netherlands
[5] Organic Geochemistry Unit, School of Chemistry, University of Bristol, Bristol, UK

*Correspondence to*: Alan T. Kennedy-Asser (alan.kennedy@bristol.ac.uk)

**Abstract.** Global and regional climate changed dramatically with the expansion of the Antarctic Ice sheet at the Eocene-Oligocene Transition (EOT). These large-scale changes are generally linked to declining atmospheric $p$CO$_2$ levels and/or changes in Southern Ocean gateways such as the Drake Passage around this time. To better understand the Southern Hemisphere regional climatic changes and the impact of glaciation on the Earth's oceans and atmosphere at the EOT, we compiled a database of 10 ocean- and 4 land- surface temperature reconstructions from a range of proxy records and compared this with a series of fully-coupled, low resolution climate model simulations from two models (HadCM3BL and FOAM). Regional patterns in the proxy records of temperature show that cooling across the EOT was less at high latitudes and greater at mid-latitudes. While certain climate model simulations show moderate-good performance at recreating the temperature patterns shown in the data before and after the EOT, in general the model simulations do not capture the absolute latitudinal temperature gradient shown by the data, being too cold particularly at high latitudes. When taking into account the absolute temperature before and after the EOT, as well as the change in temperature across it, simulations with a closed Drake Passage before and after the EOT or with an opening of the Drake Passage across the EOT perform poorly, whereas simulations with a drop in atmospheric $p$CO$_2$ in combination with ice growth generally perform better. This provides further support to previous research that changes in atmospheric $p$CO$_2$ are more likely to have been the driver of the EOT climatic changes, as opposed to opening of the Drake Passage.

## 1 Introduction

Global cooling and significant expansion of glacial ice over Antarctica at the Eocene-Oligocene Transition (EOT), ~33.7 million years ago (Ma; Zachos et al., 2001; Coxall et al., 2005), would have potentially resulted in large but uncertain changes in the Southern Ocean and the climate of the high latitude Southern Hemisphere (Bohaty et al., 2012; Passchier et al., 2013). Numerous palaeoclimate modelling studies have shown that changes in Antarctic ice sheet extent, atmospheric $p$CO$_2$ levels

and palaeogeographic reconstruction around this period of the Earth's history can all impact on the modelled global and/or regional climate (Goldner et al., 2014; Knorr & Lohmann, 2014; Kennedy et al., 2015). Interestingly, all of these studies show some areas of warming in the Southern Ocean in response to the imposition of an Antarctic ice sheet in their models, but the different models find the warming to occur in different regions. Recent modelling work using an ensemble of simulations from

the model HadCM3BL (Kennedy-Asser et al., 2019) showed that, for at least that particular climate model, the sea surface temperature response is particularly uncertain in high latitude regions due to uncertainties in the model boundary conditions that could potentially be exaggerated due to incomplete model spin-up.

While global circulation models (GCMs) are useful tools for testing our understanding of the Earth system, their inherent uncertainty within this region shows that it is necessary to integrate proxy evidence to build up a more robust picture of

Southern Ocean changes across the EOT. To this end, here we compile a large multi-proxy database of temperature for the high latitude Southern Hemisphere, incorporating a multitude of different proxy records in terms of methods, sites and temporal coverage. Despite sometimes not being directly comparable, the inclusion of very different kinds of proxy evidence provides both qualitative and quantitative measures against which model simulations can be compared and evaluated. The quantitative elements of the dataset can also be used to describe general temperature patterns (e.g. in terms of the regional mean or

latitudinal gradient), and model simulations that perform relatively well can then be used in conjunction with the proxy dataset to start to explain what changes may have occurred in this region across the EOT.

Proxy records of past climate and the 'equilibrium' climate simulations generally performed for the EOT both have strengths and weaknesses. Proxy records, specifically sediment cores, are particularly good for reconstructing the temporal domain of past climate, showing changes through long time periods at a particular point in space (e.g. Zachos et al., 2001). By contrast,

complex fully-coupled climate models generally cannot be run for long ($>10_4$ years) transient simulations and instead often only provide equilibrium snapshots of climate at a single point in time but offer a complete spatial picture of how different regions compare to one another in a physically consistent way (e.g. Goldner et al., 2014; Lunt et al., 2016; Hutchinson et al., 2018).

Equilibrium climate simulations also simplify orbital variations that would have occurred on timescales ranging from $10_4$-$10_5$ years. It is possible to take snapshots of certain points of the orbital cycle (e.g. which produce a particularly warm or cold high latitude summer) as this can be important for dictating the specific timing at which climatic thresholds might be reached, such as the point of Antarctic glaciation (DeConto & Pollard, 2003; Ladant et al., 2014). However, changing orbital parameters introduces another dimension of boundary condition variability and cannot always be sampled due to computational constraints

(e.g. Lunt et al., 2016; Kennedy-Asser et al., 2019). In proxy records, orbital variation can in some circumstances be identified (e.g. Zachos et al., 1996; Galeotti et al., 2016). However, in most cases this is either not possible due to the amount of material being required to produce a temperature estimate, which may act to average across orbital variability, or because of poorly

defined chronologies and record resolution, which could introduce uncertainty in phase relation correlation with respect to orbital variability. Most records also suffer from insufficient temporal resolution, so that at least short-scale astronomical variability (<100 kyr) is typically not clearly recovered.

The aim of this data synthesis is to create proxy datasets that are comparable to model simulations, i.e. can be used to validate the models in the spatial domain. This necessitates reducing the temporal variability of the proxy data into broad time slices, which was done for late Eocene absolute conditions (generally 36.4-34.0 Ma), relative changes across the EOT and early Oligocene absolute conditions (generally 33.2-32.0 Ma). Once time averaged, it is assumed that the records should be more representative of the longer-term climate state at their location. Dictated by the nature and inherent uncertainties in the age

models associated with the proxy data, the definition of the time slices remains reasonably crude. Indeed, proxy records used will be on different age models at each locality and cover somewhat different periods and lengths of time. This introduces an element of uncertainty. In addition to and possibly driven by orbital variability, it has been shown that there was variability in the few million years either side of the Eocene-Oligocene Boundary (E/O; e.g. Coxall & Pearson, 2007; Scher et al. 2014; Galeotti et al. 2016). Time averaging approximately 2 Ma prior to and after the E/O will however average out this temporal

variability (if a long record for a particular location is available) or potentially skew results (if for example a short-term excursion is captured in the record). To an extent these uncertainties are unavoidable and must be considered when interpreting the results presented here.

Two specific research questions are addressed in this paper. Firstly, what are the spatial patterns of temperature change inferred from proxy records for the high latitude Southern Hemisphere before, after, and across the EOT? Secondly, which GCM

boundary conditions give the best fit to the range of qualitative and quantitative proxy records of temperature before, after, and across the EOT?

A brief overview of the data synthesis, model simulation details and evaluation methods follow in Section 2. Section 3 presents the results of the model-data comparison. Finally, Sections 4 and 5 discuss the significance of the results and the potential scope of future research respectively.

## 2 Methods

### 2.1 Data synthesis

Many different proxy records for in-situ sea surface temperature (SST) are available. These include quantitative records using stable isotopes and trace metals ($\delta_{18}O$ and Mg/Ca; Bohaty et al., 2012), clumped isotopes ($\Delta_{47}$; Petersen & Schrag, 2015) and

organic biomarkers (TEX$_{86}$ and U$^{K}_{37}$; e.g. Liu et al., 2009). Quantitative proxies can be used in conjunction with qualitative records, such as nannofossil or dinoflagellate species assemblage and size (e.g. Villa et al., 2013; Houben et al., 2013), to

provide further evidence for temperature ranges or relative changes where or when quantitative data might be sparse. For example, the dinoflagellate species *S. antarctica*, broadly suggests colder temperatures with higher abundance, while its presence suggests mean annual SSTs < 10 °C (Zonneveld et al., 2013), even if spatial integration of microfossils is taken into account (Nooteboom et al. 2019).

Some terrestrial surface air temperature (SAT) records are also available, such as those derived from clay weathering products (S-index; e.g. Passchier et al., 2013) and from vegetation reconstructions (based on Nearest Living Relative, NLR, e.g. Francis et al., 2009; or the Coexistence Approach, e.g. Pound & Salzmann, 2017). These records may or may not be in-situ (in time or space), with clay weathering products for example having been exported from terrestrial regions to where they are deposited in ocean sediment cores.

Values and data are compiled from a range of sources within published material. Ideally, the data is taken from the supplementary material of the related papers. In other cases, mean values might be quoted in tables, figures or in the text of papers; however, it can be unclear over what time period these means are taken or how uncertainty values are calculated. Although this is not the most accurate way of obtaining data, in some cases this might provide the only data available and so still warrants inclusion. The sources of all data points used are outlined in detail in the supplementary information (Tables S1-

3), a digital version of which can also be accessed through the Open Science Framework (Kennedy-Asser, 2019).

These proxies respond to the climate system in different ways and all rely on various assumptions, resulting in uncertainty ranges that can be incorporated into the model-data comparison. Uncertainty in the proxy data records could arise due to calibration uncertainties or could be particularly due to temporal variability in the record (as noted in Section 1). These various

aspects make it challenging to rigorously define and quantify uncertainty. Generally, uncertainty is taken as the published values where available. Alternatively, generalised calibration uncertainty for a given proxy (if known) or two standard deviations of the temporal variability in the records can be taken as the uncertainty. Some records are presented in terms of annual temperature range and these limits can be taken as the uncertainty around the annual mean (assumed to be the mean of the maximum and minimum of the range). The source of the uncertainty ranges used are also detailed in the supplementary

material of this paper (Tables S1-3) and Kennedy-Asser (2019).

It is likely that some seasonal (summer) bias is incorporated into marine proxy records particularly at high latitudes, where light and temperature may become limiting in certain periods. In contrast, for SAT estimates based on vegetation, other conditions such as high atmospheric $p$CO$_2$ may actually push the thermal tolerances of plants to levels higher than the present-

day training set, potentially leading to (winter temperature) underestimates in reconstructions (e.g. Royer et al., 2002). Indeed, the extent of these biases is debated (Hollis et al., 2019) and may not be greater than the calibration errors that are already incorporated. Additionally, as discussed in Section 1, there could be some uncertainty in the proxy records due to variations

in the Earth's orbit. In the most extreme cases, only certain parts of orbital cycles are being captured in the sedimentary records or are strongly overrepresented. However, the sedimentary records used here, with the possible exception of the most ice-proximal sites, are generally considered to be representative of the average climate state. To quantify the effect that both of these factors might have on the temperatures around the EOT (and subsequently on the results of the model-data comparison), we vary the orbits in several model simulations and also include supplementary results showing the model-data comparison results using modelled climate averages for the summer (December, January, February) instead of annual averages. This represents the worst-case in seasonal biases and it is possible, but unlikely, that a comparable level of seasonal and orbital uncertainty exists in the proxy records. However, as there is still debate about the potential significance of these biases (Hollis et al., 2019), this uncertainty was not included in the datasets as standard. Additionally, given the long temporal averaging for each time slice, we expect orbital variability should have only a limited impact on the comparison.

Some studies (e.g. Waelbroeck et al., 2009; Dowsett et al., 2012; Pound & Salzmann, 2017) devise semi-quantitative metrics for the quality of proxy records, based upon factors such as preservation, dating quality, calibration errors etc. when compiling their datasets. Here, there is no formal assessment of the quality of individual proxies or records, nor is there any reinterpretation or recalculation of existing datasets, as this would beyond the scope of the paper. Instead, here the dataset integrates as many independent proxies as possible for each site, and all are used to evaluate the model simulations. It is important to note that the same proxy is only used in the compilation once per site per time slice. If two or more records using the same proxy at the same site are available, generally the most recent value in the literature is used (e.g. Passchier et al., 2013 and 2016 both provide estimates for temperature using the S-index in Prydz Bay, so the 2016 value is used). Different proxies are weighted equally in the model evaluation, with sites where there are multiple records therefore being weighted more strongly for the purpose of model-data comparison.

In total, data were taken from 14 sites (10 ocean and 4 terrestrial), ranging in palaeolatitude from 53 to 77 °S and palaeolongitude from 63 °W to 177 °E. The compiled temperature records are shown for the late Eocene and for the early Oligocene in Figure 1, and for the change across the EOT in Figure 2. The references for all data points are included in the supplementary information.

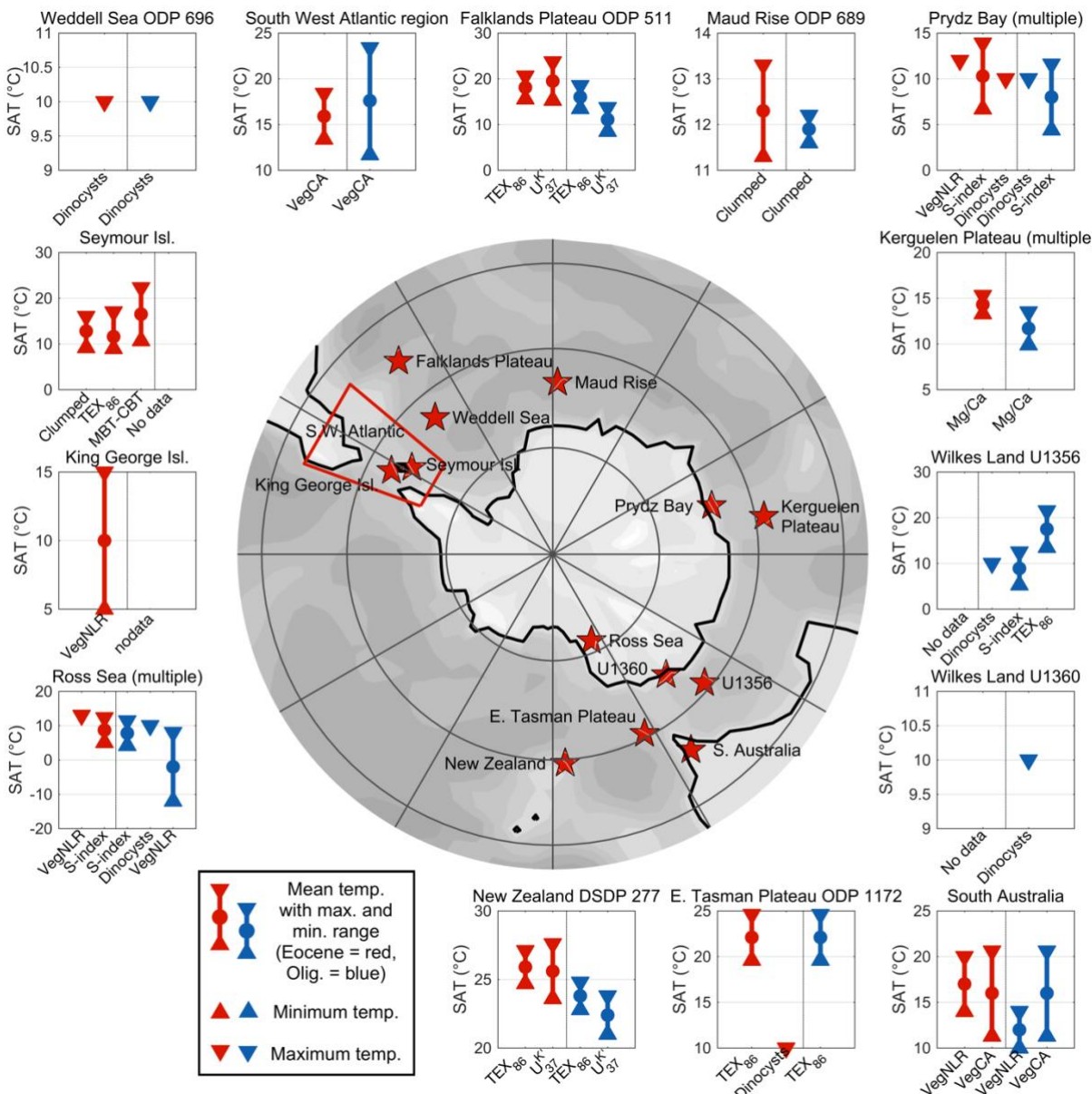

**Figure 1: Mean annual temperature (°C) from proxy records for all sites during the late Eocene and early Oligocene. The mean values (circles) are shown with maximum and minimum values (error bars), while ordinal limits are shown by upwards (greater than) or downwards (less than) pointing triangles. Late Eocene records are in red and early Oligocene records in blue.**

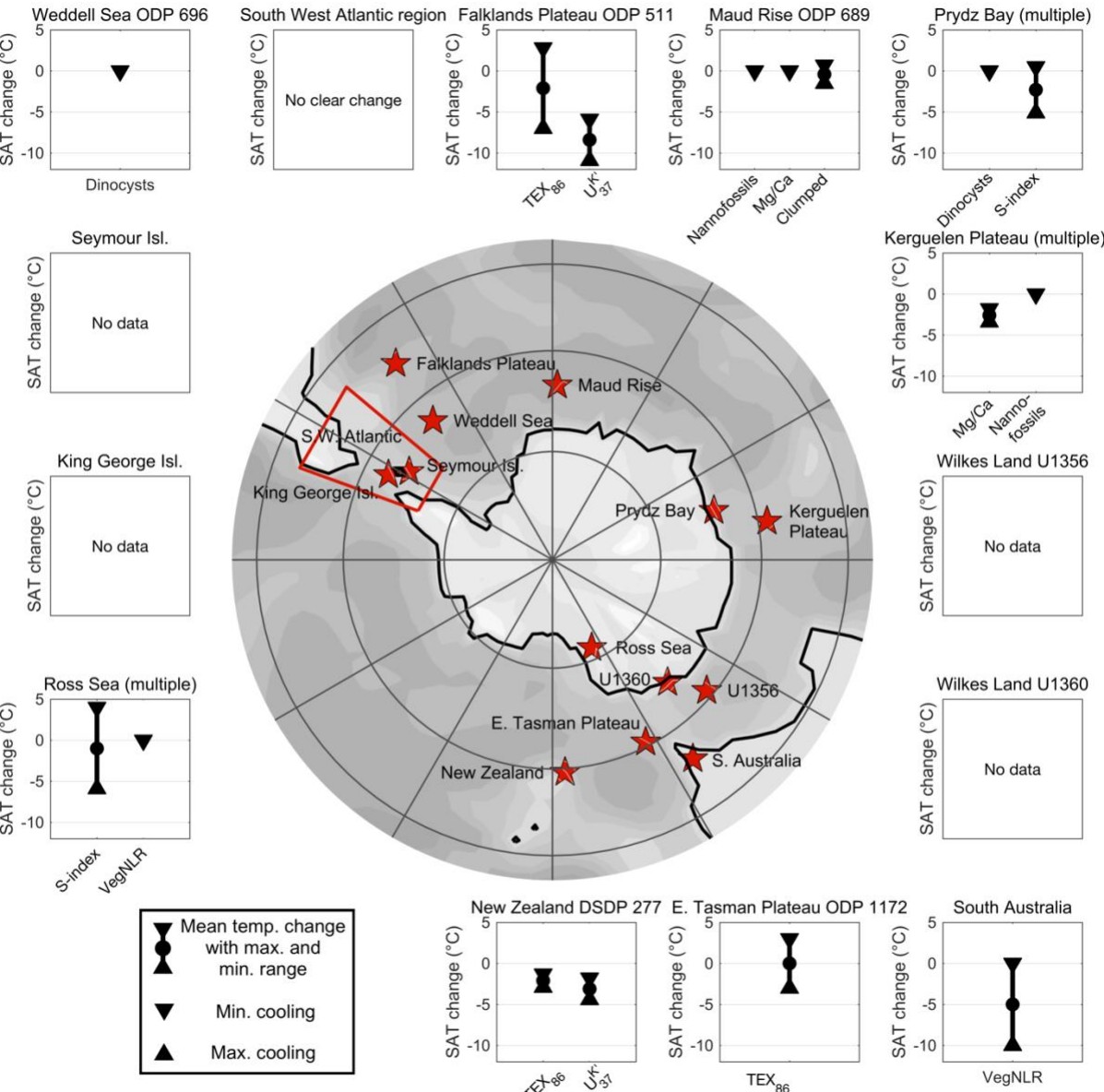

**Figure 2: Changes in mean annual temperature (°C) from proxy records for all sites across the EOT. The mean values (circles) are shown with maximum and minimum values (error bars), while ordinal limits are shown by upwards or downwards pointing triangles.**

**2.2 Model simulations**

The proxy datasets compiled here are compared to the fully spun-up HadCM3BL-M2.1aE (HadCM3BL henceforth) simulations outlined in Kennedy-Asser et al. (2019) and simulations from FOAM outlined in Ladant et al. (2014). An overview of the simulations used is provided in Table 1. A detailed description of the models' setup and the simulation details can be found in the respective references.

**Table 1: Brief overview of climate models and the boundary conditions varied for each.**

| Model | Atmos. resolution | Ocean resolution | No. of simulations used | Simulation length (years) | Palaeogeog. vars. | Ice sheet vars. | $p$CO$_2$ vars. (ppmv) | Reference |
|---|---|---|---|---|---|---|---|---|
| HadCM3BL-M2.1aE | 96x73x19 (3.75 x 2.5 °) | 96x73x20 (3.75 x 2.5 °) | 8 | >6,000 | Open Drake Passage (1,111 km wide) | No ice | 840 560 | Kennedy-Asser et al., 2019* |
| | | | | | | EAIS | 840 560 | |
| | | | | | Closed Drake Passage | No ice | 840 560 | |
| | | | | | | EAIS | 840 560 | |
| FOAM | 48x40x18 (7.5 x 4.5 °C) | 128x128x24 (~2.8 x 1.4 °C) | 12 (6 x warm orbit; 6 x cold orbit) | 2,000 | Open Drake Passage (1,250 km wide) | No ice | 1,120 840 560 | Ladant et al., 2014 |
| | | | | | | Small EAIS | 560 | |
| | | | | | | EAIS | 560 | |
| | | | | | | Full AIS | 560 | |

*HadCM3BL simulations are those from the 'Spin-up ensemble' in Kennedy-Asser et al. (2019), which were selected as they are more adequately spun-up. These have a present-day orbital configuration.*

These models are all relatively low resolution and less complex than some others that have been used in recent studies (e.g. Hutchinson et al., 2018; Baatsen et al., 2018); however, they are still regularly used in palaeoclimate research (e.g. Goddéris et al., 2017; Farnsworth et al., 2019; Saupe et al., 2019). For the present day, HadCM3BL is shown to perform comparably to CMIP5 models in terms of a number of global mean variables, although it produces a moderate cold bias globally, with high (northern) latitudes being too cold because of an exaggerated seasonal cycle and overly cold winter (Valdes et al., 2017). This bias is similar to other higher resolution variants of the model (Valdes et al., 2017). FOAM has been shown to capture most of

major characteristics of present-day climatology (Jacob, 1997; Liu et al., 2003) as well as reasonable climate variability (Wu & Liu, 2005). As HadCM3BL, FOAM exhibits a cold high latitude bias in the Northern Hemisphere, in particular in winter (Gallimore et al., 2005).

It should be noted that for both models the Antarctic ice sheets are prescribed and cannot expand or contract through the simulations, and also that, along with the palaeogeographies, the ice sheets used are different between the original studies. Orbital variability was accounted for in the FOAM simulations (Ladant et al., 2014), having both a warm summer and cold summer orbital variants available for comparison. Given that it is not possible to definitively show if proxy records are capturing extreme cases of orbital variability, these simulations are used to inform the potential magnitude of uncertainty this

might introduce. The model spin-up period also differs between the two studies, with the HadCM3BL simulations being significantly longer. The HadCM3BL simulations were selected from Kennedy-Asser et al. (2019) based upon their extended spin-up, meaning the modelled results are expected to be highly robust with negligible trends to bias the conclusions. FOAM simulations have been integrated for 2,000 years and are in equilibrium in the upper ocean. Small cooling trends exist in the deep ocean but the rates of temperature change are smaller than 0.1°C/century, which is a criterion regularly used to define

quasi-equilibrium (e.g., Lunt et al. 2017). However, it if possible that there could be some model drift if the simulations were run out beyond 2,000 years, depending on the initial condition (Farnsworth et al, 2019).

In order to evaluate against the proxy dataset of relative changes across the EOT, pairs of model simulations can be selected that represent the forcing changes occurring across the EOT. These pairs of model simulations represent a before and an after

state, with the difference in the boundary conditions between the pairs described as the forcing and the difference in the modelled climate representing the change across the EOT. Given that the vast majority of glaciological proxy data gives evidence of glacial expansion, here the modelled forcing must include some sort of ice expansion (i.e. the early Oligocene simulation must contain an ice sheet and the late Eocene simulation must contain no ice sheet). The simulation pairs may additionally include other forcing changes that are potentially relevant to describe the state of the Earth system before and after

the EOT, namely $pCO_2$ level and gateway configuration. Simulation pairs were chosen that represented:

- An expansion of ice over Antarctica from an ice-free state to either an EAIS of full AIS, with all other boundary conditions remaining the same
- A similar expansion of ice over Antarctica but also combined with a simultaneous drop in $pCO_2$, with palaeogeographic boundary conditions remaining the same

- A similar expansion of ice over Antarctica but also combined with a simultaneous change in palaeogeography (an opening of the Drake Passage), with $pCO_2$ boundary conditions remaining the same
- A similar expansion of ice over Antarctica but also combined with a simultaneous change in palaeogeography (an opening of the Drake Passage) and a drop in $pCO_2$

This produced 9 pairs of simulations from HadCM3BL and 18 pairs from FOAM. FOAM simulations were always compared with the same orbital variability before and after the EOT. A detailed description of all simulations and simulation pairs used is included in Table S4 of the supplementary material.

## 2.3 Metrics of comparison

Most proxies in this compilation provide continuous quantitative data that can be directly compared to models or other records, (e.g. absolute temperature estimates from geochemical proxies). Other proxies may provide ordinal (qualitative) data; that is, data that can be ranked into an order of greater or lesser magnitude but from which absolute values are not attainable (e.g. dinoflagellate species assemblage). Both of these kinds of data can be used to evaluate the palaeoclimate model simulations.

At each site where proxy data is available, the modelled annual mean air or water temperature is taken as the mean over a three by three grid cell area surrounding each proxy location, with the maximum and minimum modelled temperature also taken from these nine grid cells as the modelled uncertainty. Given the relatively coarse resolution of these models, this represents a very large area (ranging 2.25-6.35 × 10$_5$ km$_2$). This method will therefore only capture large scale climate variability and not local variations. The principal method used to evaluate the GCMs against the proxy dataset is the root mean square error (RMSE), which simply finds the mean difference between the models and the data for all comparable points. The RMSE is calculated in two ways:

Firstly, the 'standard' RMSE, defined in Eq. (1), is calculated from the maximum or minimum of the uncertainty range of the proxy data to the minimum or maximum of the uncertainty range in the model (if the model is too warm or cold, respectively).

$$standard\ RMSE = \sqrt{\frac{\sum_n^i E_{S,i}{}^2}{n}}, \tag{1}$$

Where, $E_S$ is the error, defined in Eq. (2), and $n$ is the number of proxy records for a given time slice.

$$E_{S,i} = T_{p,i} - T_{m,i}, \tag{2}$$

Where $T_p$ is the range of temperatures indicated by proxy reconstruction $i$ and $T_m$ is the range of temperatures indicated by a model simulation for the location of record $i$. The standard error, $E_s$, is taken as zero if the range of model uncertainty, $T_m$, overlaps the range of proxy uncertainty, $T_p$. This can be calculated for continuous data or ordinal data that provides an upper range for the temperature, such as the presence of *S. antarctica*. Examples of how this is applied are illustrated in Supplementary Figure 1.

Secondly, the RMSE is calculated once the mean temperature of all data points/sites (either in the proxy dataset or for a given model simulation) has been removed. The purpose of removing the mean is so the model performance is not primarily judged against systematic warm or cold biases, the latter of which are typical at high latitudes in palaeoclimate simulations of past warm climates (Huber & Caballero, 2011; Lunt et al., 2012). This 'normalised' RMSE, defined in Eq. (3), instead evaluates the spatial pattern of temperature in the Southern Ocean. This metric is used with continuous data where a mean value is available, again with the error taken between the ranges of the proxy and model uncertainty.

$$normalised \ RMSE = \sqrt{\frac{\sum_n^i E_{N,i}^2}{n}} \ , \tag{3}$$

Where, $E_N$ is the error of the normalised data, defined in Eq. (4).

$$E_{N,i} = (T_{p,i} - \overline{T_p}) - (T_{m,i} - \overline{T_m}) \ , \tag{4}$$

Where $\overline{T_p}$ is the mean temperature of all proxy records and $\overline{T_m}$ is the mean modelled temperature across all proxy record sites.

'Count metrics' can also be used for the absolute and relative change data comparisons, allowing a large range of proxy records to be incorporated. These metrics count how many of the data points the model is consistent with in terms of magnitude (i.e. within the error bars of.) and, for the change across the EOT, the number of records for which the model simulations correctly predict the direction of change. This can allow ordinal data (such as increasing cold water taxa) to contribute to the comparison.

In order to assess the simulations across multiple criteria, metric scores that have comparable units (e.g. the two RMSE metrics) can simply be summed or averaged. Additionally, to further expand upon the idealised model-data comparison of Kennedy-Asser et al. (2019), it is important to consider not just if the simulated change across the EOT is realistic, but also if the starting and ending state are realistic compared to the late Eocene and early Oligocene datasets. This is done by combining the metric scores for a pair of simulations that describe the change across the EOT (compared to the EOT dataset) with metric scores for the pre- and post-EOT simulations that make up that pairing (compared to the late Eocene and early Oligocene datasets). If the datasets had a consistent spatial coverage for each of the time slices, the difference between the late Eocene and early Oligocene absolute datasets would be the same as the EOT relative change dataset. However, because there are some sites with records available only before or after the EOT, and some relative changes for which absolute values are not available, the pair of simulations that gives the best fit before and after the EOT is not necessarily the same pair that gives the best fit for the observed change across the EOT. Which metric is used to evaluate across the time slices and if there is any weighting put on the absolute or relative change datasets is subjective. Although the count metrics are shown for reference, here, the combined rank score for each time slice is based upon only the two RMSE metrics and the three time slices (late Eocene, early Oligocene and EOT) are weighted equally.

**2.4 Benchmarks for evaluation**

For the model simulations to be described as performing particularly 'well' or 'poorly', it is necessary to have some sort of benchmark to compare the models' performance against. For the three time slices, two benchmarks are used: these can be thought of as hypothetical generalisations of the whole regional high latitude Southern Hemisphere climate based only upon proxy data. First, the mean temperature (or temperature change) of all sites and proxies is taken as a homogeneous value at all sites. Second, the ordinary least squares linear fit through the mean temperatures (or temperature change) with palaeolatitude from all proxies and sites, shown in Figure 3, is taken to produce a synthetic, latitudinally varying temperature field for the region. If model simulations perform better than both benchmarks, they can be described as showing *good* performance as they are correctly modelling zonal and regional variation beyond this general latitudinal trend. If the simulations perform worse than both benchmarks, they show *poor* performance and are failing to identify even the most basic variation in the dataset. If the simulations outperform the constant mean benchmark but not the latitudinal gradient benchmark, they are described as showing *moderate* performance. When evaluating the model simulations across both RMSE metrics, if a simulation outperforms a benchmark for one metric but not the other, its performance can be described for example as *moderate-poor*.

## 3    Model-data comparison

**3.1 Latitudinal temperature profiles**

The regional mean of the proxy records and latitudinal temperature gradient benchmarks are shown in Figure 3 along with the best HadCM3BL and FOAM simulations identified in Section 3.5. In addition to the latitudinal gradient calculated for the full dataset for each time slice, an uncertainty range for the gradient was calculated by systematically omitting single points from the regression to test for potential bias in the proxy record compilation. The absolute temperature profiles in the late Eocene and early Oligocene proxy datasets show colder temperatures at higher latitudes than mid-latitudes, as would be expected. The latitudinal gradient is comparable between the early Oligocene (0.54 °C °N$_{-1}$; range 0.30 to 0.63 °C °N$_{-1}$) and the late Eocene (0.49 °C °N$_{-1}$; range 0.45 to 0.54 °C °N$_{-1}$), with the Oligocene gradient more uncertain due to the greater variability in the proxy records. The change in temperature across the EOT identified by the proxies has a negative slope dataset (-0.20 °C °N$_{-1}$; range -0.34 to -0.11 °C °N$_{-1}$), suggesting that cooling is greater at mid-latitudes and is less at higher latitude sites, although the steepness of the gradient for the EOT is somewhat enhanced by the strong cooling at the lower latitude Falklands Plateau. Implications of this latitudinal gradient change across the EOT will be discussed further in Section 4.2.

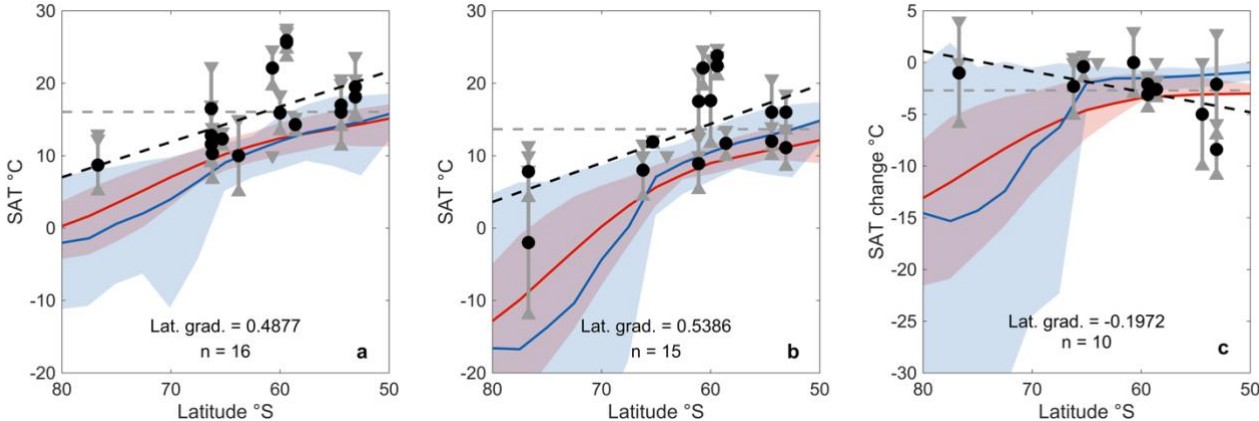

**Figure 3: Latitudinal profiles of a) late Eocene absolute temperature, b) early Oligocene absolute temperature and c) EOT temperature change from proxy records. The regional mean values are plotted in grey dotted lines and latitudinal gradients (calculated using ordinary least squares) in black dotted lines. Circles show proxy data mean values, while their uncertainty ranges and maximum/minimum limits are shown by the bars and triangles. The coloured lines show the zonal mean surface air temperature profile for the best HadCM3L (blue) and FOAM (red) simulations, with the shading showing the zonal maximum and minimum surface air temperature for each model.**

## 3.2 Late Eocene temperatures

The standard RMSE, normalised RMSE and count metric for all of the ice-free simulations and the benchmarks in comparison to the late Eocene dataset are shown in Figure 4a for the annual mean temperature. Equivalent simulations for the summer mean temperatures are shown in Supplementary Figure 2a. The standard RMSE scores show that absolute temperature biases are in general large compared with the benchmarks. The standard RMSE scores are better for simulations at higher $pCO_2$ levels for both HadCM3BL and FOAM, showing there is a cold bias in the simulations from both models, consistent with issues faced in previous research of this period (Lunt et al., 2012). This is consistent across all sites, although the fit with the New Zealand records is particularly poor. As a result, only one HadCM3BL simulation outperforms the homogeneous benchmark (3x pre-industrial $pCO_2$ levels and an open Drake Passage). No simulations from either model outperform the latitudinal gradient benchmark in terms of the standard RMSE. FOAM simulations with a colder summer orbit actually produce a slightly better fit with the data compared to the alternative warmer summer orbit, but they are still significantly higher than either benchmark. Modelled summer temperatures (Supplementary Figure 2a) give a better fit for the standard RMSE, with four simulations having *moderate* performance, however still no simulations can be described as *good*.

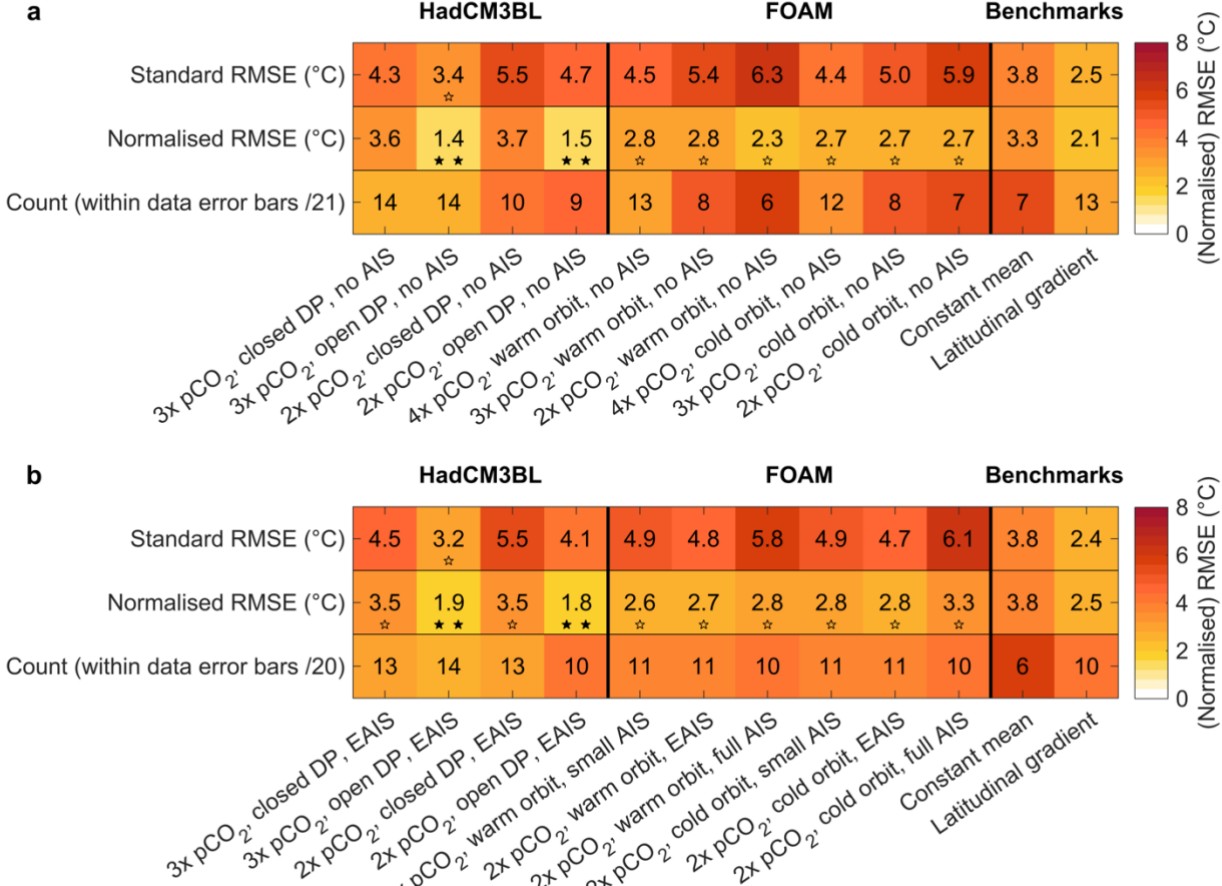

**Figure 4: Standard RMSE (°C), normalised RMSE (°C) and count metric for the annual mean temperature from all model simulations and the benchmarks compared against the late Eocene dataset (a) and the early Oligocene dataset (b). Labels on the x-axis refer to the $pCO_2$ level ('2x', '3x' or '4x' pre-industrial levels), state of the Drake Passage ('DP') in HadCM3BL simulations and the size of the Antarctic ice sheet (AIS). The colour scale of the count metric is** 5 **normalised to match that of the RMSE metrics (i.e. white = best: all sites are within error bars; dark orange = worst: no sites are within error bars). For a given metric, single open stars indicate simulations with *moderate* performance and double black stars indicate simulations with *good* performance.**

When the mean temperature bias is removed for the normalised RMSE, more of the simulations outperform the constant mean benchmark and some outperform the latitudinal gradient benchmark. For FOAM, with the cold temperature bias removed, the

10 lower $pCO_2$ simulation with a warm orbit performs better than the higher $pCO_2$ simulations with a warm orbit and all outperform the constant mean benchmark. The best simulations are those from HadCM3BL with an open Drake Passage, which perform better than the latitudinal gradient benchmark. The worst simulations are the HadCM3BL simulations with a

closed Drake Passage, both of which fail to outperform either benchmark, suggesting this palaeogeographic configuration has a major influence on the spatial patterns of temperature and is unrealistic. For the normalised RMSE, summer temperatures perform worse than the annual mean for all simulations.

No simulations outperform both benchmarks for both RMSE metrics, so none can be described as *good* by our definition. However, the HadCM3BL simulation at 3x PI $p$CO$_2$ with an open Drake Passage outperforms the homogenous benchmark for both metrics and the latitudinal gradient benchmark for the normalised RMSE, so can be described as *moderate-good*.

### 3.3 Early Oligocene temperatures

Figure 4b shows the standard RMSE, normalised RMSE and count metric for all glaciated simulations against the early Oligocene dataset for the annual mean temperature (with summer temperatures shown in Supplementary Figure 2b). Again, there is a general cold bias indicated by the poorer standard RMSE scores for the lower $p$CO$_2$ simulations for HadCM3BL. The New Zealand records are poorly represented again by all models, but there are also issues representing the Maud Rise, East Tasman Plateau and one of the Ross Sea records. Generally, the standard RMSE values are similar to the late Eocene comparison. Again, only one simulation outperforms the constant mean benchmark: HadCM3BL at 3x pre-industrial $p$CO$_2$ with an open Drake Passage, with no simulations outperforming the latitudinal gradient benchmark for this metric. The FOAM simulations with the largest ice sheet configurations have poorer standard RMSEs compared to the FOAM simulations with smaller ice sheets, likely due to the cooling (and hence cold bias) being greater with a larger ice sheet. The differences in orbits have little effect on the performance of the FOAM simulations. Similarly to the late Eocene, modelled summer temperatures (Supplementary Figure 2b) fit the data better, with all FOAM simulations and both HadCM3BL simulations with an open Drake Passage showing *moderate* performance.

For the normalised RMSE, all simulations outperform at least one benchmark. The HadCM3BL simulations with an open Drake Passage at either $p$CO$_2$ level are the joint best. Again, as with the late Eocene temperature data, the HadCM3BL simulations with the closed Drake Passage perform much worse than the equivalent open Drake Passage simulations in terms of the both RMSE metrics. For this metric, the FOAM simulations with the largest AIS do not perform as well as those with smaller ice sheet configurations (although the difference is not so marked) and the different orbits have little effect. This could suggest the AIS expansion across the EOT might not be at the upper range of volume estimates suggested by other studies (e.g. Bohaty et al., 2012; Wilson et al., 2013), but it should be noted that the maximum ice extent is likely lost in the time averaging, even if present in the records. Either way, this result should be treated with caution, as although summer temperatures generally result in worse normalised RMSE scores for all simulations, notable exceptions are the FOAM simulations with the largest AIS, which show a slight improvement (Supplementary Figure 2b).

Like for the late Eocene, no simulation can be described as *good* for both RMSE metrics; however, the glaciated HadCM3BL simulation at 3x pre-industrial $p$CO$_2$ with an open Drake Passage can be described as *moderate-good*.

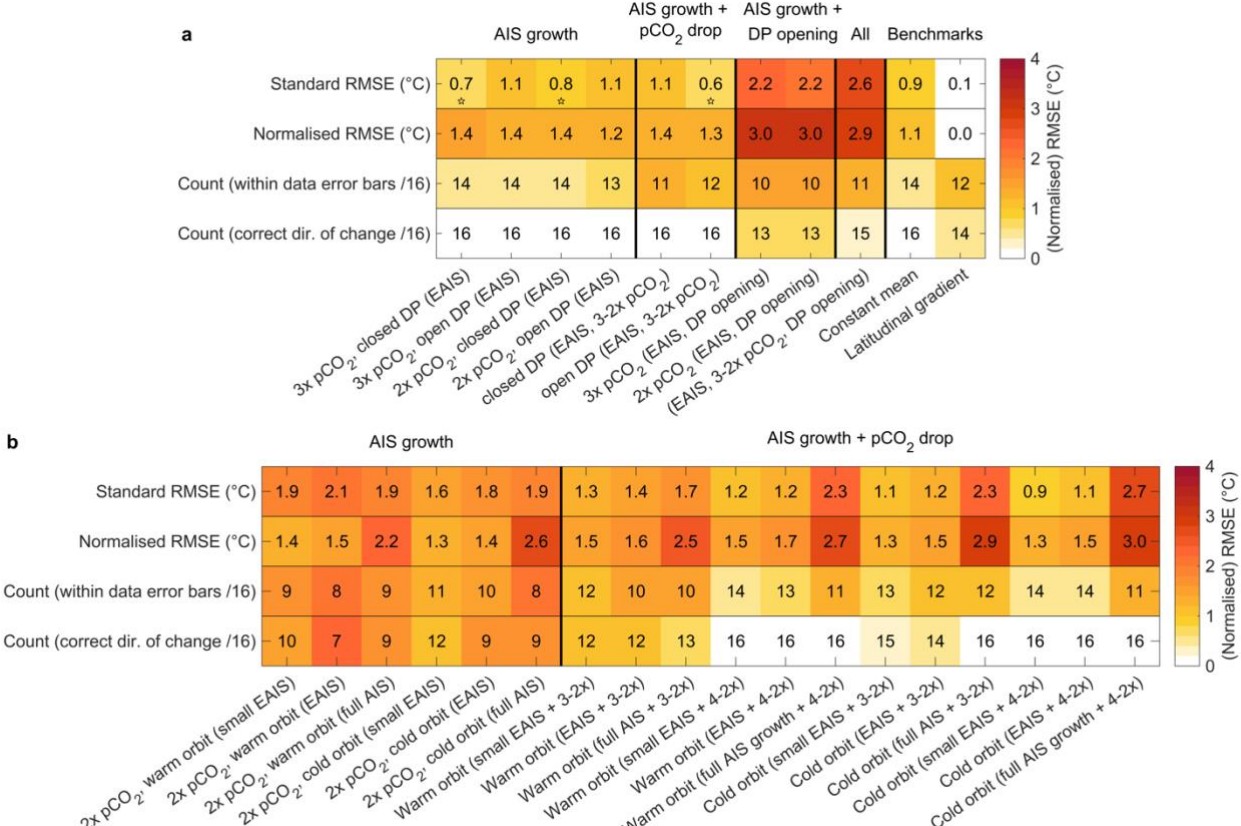

**Figure 5: Standard RMSE (°C), normalised RMSE (°C) and count metrics for the annual mean temperature from all pairs of model simulations representing the forcing across the EOT compared against the EOT dataset from HadCM3BL and the benchmarks (a) and FOAM (b). The simulation pairs are grouped by forcing. Labels on the x-axis are similar to Figure 4, with changes in boundary conditions associated with each pair of simulations written in brackets. The colour scales of the count metrics are normalised to match that of the RMSE metrics and stars indicate *moderate/good* performance as in Figure 4.**

### 3.4 EOT temperature change

All pairs of model simulations representing the change in annual mean temperature that occurred across the EOT are shown in Figure 5. A comparable plot using summer mean temperatures is shown in Supplementary Figure 3. It is important to note that, generally, the uncertainties in the EOT dataset are much greater relative to the magnitude of change, compared to the uncertainties relative to the absolute values in the late Eocene and early Oligocene datasets. As a result, the latitudinal gradient benchmark provides a remarkably good fit for the data covering the EOT, lying almost entirely within the data uncertainty. No

model simulations perform as well as this benchmark, but again, because the uncertainty in the change relative to its magnitude is greater than in the absolute datasets, generally the model RMSE scores are lower for this dataset than the late Eocene or early Oligocene datasets. For this dataset, there is not a clear picture of modelled changes being over- or underestimated relative to the proxy records. The largest error for all models is in representing the large cooling shown at the Falklands Plateau.

Three HadCM3BL simulation pairs (Figure 5a) outperform the constant mean change benchmark for the standard RMSE metric: those with an open Drake Passage in response to AIS growth and a $pCO_2$ drop and those with a closed Drake Passage in response to AIS growth at both $pCO_2$ levels. No simulation pairs outperform either benchmark for the normalised RMSE metric. In contrast to what was shown for the absolute temperature dataset comparisons for the late Eocene and early Oligocene, the HadCM3BL simulation pairs with a closed Drake Passage (both before and after the EOT) perform relatively well,

particularly for the standard RMSE. This shows that although simulations can be far from the proxies in absolute terms, they can still produce promising results in other ways.

Similar to what was shown for the late Oligocene, the FOAM simulations (Figure 5b) generally fit the dataset best in terms of the standard and normalised RMSE when they have smaller ice sheets added. Although it makes little difference for the normalised RMSE scores, FOAM simulations which combine a $pCO_2$ drop in tandem with AIS growth perform better in terms

of the standard RMSE and the count metrics (i.e. the number of sites which lie within error bars or simulate the correct direction of change) than those with which simulate only AIS growth. The orbital variations make little difference to the FOAM performance, with simulations with small ice sheets performing fractionally better with a colder orbit and those with a larger ice sheet performing slightly better with a warmer orbit.

Generally in terms of the forcings across all model simulation pairs, the AIS growth forcing in isolation produces the best

normalised RMSE and performs comparably to the combined AIS growth and $pCO_2$ drop forcing in terms of the standard RMSE. For HadCM3BL, the AIS growth forcing produces better results for the count metric of sites within the data error bars whereas for FOAM the combined AIS growth and $pCO_2$ drop forcing produces better results for the count metrics. The HadCM3BL simulations with an opening of the Drake Passage (in combination with AIS growth or AIS growth and $pCO_2$ drop) generally gives the poorest fit for the RMSE metrics of all the simulation pairs and the worst count metric results of all

HadCM3BL simulation pairs. This suggests that opening of the Drake Passage across the EOT is the least likely of these model scenarios.

No simulations from any model perform better than either benchmark for both RMSE metrics, with the best HadCM3BL simulation pairing (with an open Drake Passage in response to both AIS growth and $pCO_2$ drop) coming the closest (its normalised RMSE being 0.24 °C worse than the constant mean benchmark). All simulation pairs can therefore only be

described as *moderate-poor* or *poor*. Using modelled summer temperatures (Supplementary Figure 3), generally produces a worse fit to the EOT dataset for both models with all model simulation pairs having *poor* performance. The reasoning for this poor performance is discussed further in Section 4.

## 3.5 Evaluation across time slices

As noted in Section 2.3, it is possible to evaluate the model simulations and model simulation pairs across various metrics. The best five simulations (or simulation pairs) for the late Eocene, early Oligocene and for the change across the EOT based on the mean of their two RMSE metrics are shown in Table 2, along with the mean of the two RMSE metrics for each of the benchmarks for comparison. As well as taking the average RMSE for each time slice, the average RMSE can be taken across all three time slices. It is not always the case that simulation pairs that perform well for the observed EOT change also perform well when the late Eocene and early Oligocene data are incorporated. As was noted in Sections 3.1 and 3.2, for the absolute temperatures, simulations with a closed Drake Passage perform relatively poorly. As a result, when the combined ranked performance score is calculated across all three time slices, the pairings with a closed Drake Passage are not found to perform as well, highlighting the importance of incorporating the absolute values into this model-data comparison. Again, this suggests that the Drake Passage was open prior to the EOT and the late Eocene. The best five simulations in terms of the mean standard RMSE and normalised RMSE across all three time slices are also listed in Table 2, along with the benchmarks for comparison.

**Table 2: The five highest ranked simulations (or simulation pairs) in terms of mean standard and normalised RMSE for each time slice and across all three time slices.**

| Rank | Mean RMSE (°C) | Model | $pCO_2$ (ppmv) | AIS state | Drake Passage | Orbit |
|---|---|---|---|---|---|---|
| | | | *Late Eocene absolute* | | | |
| | 2.28 | | Latitudinal gradient benchmark | | | |
| 1 | 2.42 | HadCM3BL | 840 | No ice | Open | Normal |
| 2 | 3.08 | HadCM3BL | 560 | No ice | Open | Normal |
| 3 | 3.54 | FOAM | 1,120 | No ice | Open | Cold orbit |
| | 3.56 | | Constant mean benchmark | | | |
| 4 | 3.64 | FOAM | 1,120 | No ice | Open | Warm orbit |
| 5 | 3.85 | FOAM | 840 | No ice | Open | Cold orbit |
| | | | *Early Oligocene absolute* | | | |
| | 2.51 | | Latitudinal gradient benchmark | | | |
| 1 | 2.54 | HadCM3BL | 840 | EAIS | Open | Normal |
| 2 | 2.99 | HadCM3BL | 560 | EAIS | Open | Normal |
| 3 | 3.72 | FOAM | 560 | EAIS | Open | Cold orbit |
| 4 | 3.74 | FOAM | 560 | EAIS | Open | Warm orbit |
| 5 | 3.77 | FOAM | 560 | Small EAIS | Open | Warm orbit |
| | 3.79 | | Constant mean benchmark | | | |
| | | | *EOT change* | | | |
| | 0.05 | | Latitudinal gradient benchmark | | | |
| | 0.97 | | Constant mean benchmark | | | |
| 1 | 0.97 | HadCM3BL | 840 – 560 | No ice – EAIS | Open – Open | Normal |
| 2 | 1.08 | HadCM3BL | 840 – 840 | No ice – EAIS | Closed – Closed | Normal |
| 3 | 1.10 | HadCM3BL | 560 – 560 | No ice – EAIS | Closed – Closed | Normal |
| 4 | 1.14 | FOAM | 1,120 – 560 | No ice – small EAIS | Open – Open | Cold orbit |
| 5 | 1.17 | HadCM3BL | 560 – 560 | No ice – EAIS | Open – Open | Normal |
| | | | *Late Eocene absolute + EOT change + Early Oligocene absolute* | | | |
| | 1.61 | | Latitudinal gradient benchmark | | | |
| 1 | 2.06 | HadCM3BL | 840 – 840 | No ice – EAIS | Open – Open | Normal |
| 2 | 2.13 | HadCM3BL | 840 – 560 | No ice – EAIS | Open – Open | Normal |
| 3 | 2.41 | HadCM3BL | 560 – 560 | No ice – EAIS | Open – Open | Normal |
| | 2.77 | | Constant mean benchmark | | | |
| 4 | 2.84 | FOAM | 1,120 – 560 | No ice – EAIS | Open – Open | Cold orbit |
| 5 | 2.86 | FOAM | 1,120 – 560 | No ice – small EAIS | Open – Open | Cold orbit |

## 4 Discussion

### 4.1 Plausible forcings of EOT climatic change

This model-data comparison shows that the most realistic representation of the high latitude Southern Hemisphere climate before, after and across the EOT would be simulated by an expansion of an AIS, possibly with some combination of atmospheric $pCO_2$ decline. Despite limitations in the modelled absolute temperature before and after the EOT, incorporating this information into the comparison influences which simulation pairs are identified as best at representing how the climate might have changed across the EOT. Without accounting for the absolute data, simulation pairs with a closed Drake Passage can perform well, whereas for the absolute data these simulations perform poorly.

The marked reduction in performance by HadCM3BL when the Drake Passage either is closed before and after the EOT or is closed before but opens across the EOT supports the conclusions of Goldner et al. (2014) that changes in ocean gateways around the EOT are not the best way to model the changes observed in the proxy record. This is in support of the general shift in consensus away from the gateway hypothesis as the sole cause of the changes at the EOT, at least in terms of the direct climatic implications (DeConto & Pollard, 2003; Huber & Nof, 2006; Sijp et al., 2011; Ladant et al., 2014 etc.). However, a pre-conditioning by gateway deepening and invigorated Antarctic Circumpolar Current is still plausible based on SST proxy data and microfossil distribution from directly prior the EOT (Houben et al., 2019). There is inconclusive evidence in the literature for fundamental changes in the Drake Passage around the EOT (e.g. Lagabrielle et al., 2009 and references therein), in agreement with the Getech palaeogeographic reconstructions, which have the gateway open throughout the period (see the Lunt et al., 2016, Figure S1; Kennedy-Asser et al., 2019, Supplementary Figure 1). However, it should be noted that proxy evidence and reconstructions suggest the Tasman Seaway deepened close to, but probably prior to the EOT (e.g. Stickley et al., 2004; Scher et al., 2015, Houben et al., 2019) and this could have different implications on the climate (which are potentially more consistent with temperature proxy records compiled here) from the results shown for Drake Passage opening. The preconditioning effects of widening and deepening the Tasman Seaway therefore could be of interest to focus on with future model comparisons.

It is important to bear in mind that this result was obtained from a relatively low-resolution model. With higher resolution models, it is possible that changes in modelled ocean circulation and atmospheric response could be very different, particularly given that much smaller changes in the Southern Ocean gateways than were modelled here could have occurred across the EOT (e.g. Viebahn et al., 2016) For this paper, it was not feasible to use higher resolution models for such a range of boundary conditions and length of simulations, and this should remain an important priority in future research.

The better fit with proxy data by FOAM when the AIS is not at its full extent also would be consistent with the other glaciological evidence. Various sites around the Ross Sea showed the maximum AIS expansion occurring around ~32 Ma (e.g. Olivetti et al., 2015; Galeotti et al., 2016), significantly after the EOT, while sedimentological evidence from the Weddell Sea suggests that region of West Antarctica was not fully glaciated until much more recently (~15 Ma; Huang et al., 2014). If this

climatic fingerprint of a smaller AIS is robust, given that this signal already appears to be present in the data even with only limited site locations, there could be potential in future work to be able to constrain the extent of the AIS using only a climate model and proxy temperature records, which could then be used to independently verify other estimates from ice sheet modelling or proxy estimates using $\delta_{18}O$.

HadCM3BL simulations with differing AIS extent boundary conditions (those with the Getech palaeogeographic reconstructions from Kennedy-Asser et al., 2019) also show a similar result, with simulations with a smaller EAIS fitting the data better (figure not shown). However, as was discussed in Kennedy-Asser et al., these simulations are potentially not fully spun-up and so they are not included in the analysis of this paper. It should be noted that the FOAM simulations have a relatively short spin-up of 2,000 years (Table 1) and without deeper investigation into the time series of the model spin-up, it
is not possible to say if this model is yet fully in equilibrium.

## 4.2 Discrepancies and uncertainty in the latitudinal temperature gradient

Although this model-data comparison provides some interesting results, there is still clear room to improve the models' performance and reduce discrepancies with the data. The zonal mean temperature for each of the best pairs of simulations from
HadCM3BL and FOAM (across all three time slices) are shown in Figure 3 along with the proxy records and their uncertainty. For the late Eocene and early Oligocene, the latitudinal gradients produced by the models are reasonably similar to the gradients shown in the proxy records, although the models generally have a cold bias of around 5-10 °C (Figures 3a and 3b). The models provide a better representation of the relative spatial patterns of temperature (i.e. for the normalised RMSE metric) compared to the absolute temperatures (i.e. for the standard RMSE metric) because of this systematic cold bias at high latitudes.

Although there could be an element of seasonal bias in some of the proxy records (Hollis et al. 2019) that could explain absolute temperature biases before and after the EOT, the supplementary results presented here find that using modelled summer temperatures generally results in worse model performance for the relative change across the EOT and for the normalised RMSE scores. Higher resolution modelling and better representation of climate feedbacks offer some potential improvements in this regard (Huber & Caballero, 2011; Baatsen et al., 2018) and the current DeepMIP modelling effort (Lunt et al., 2017)
might provide further insights into the causes of this common model bias. It should also be noted that these simulations were run with relatively arbitrary $p\text{CO}_2$ levels (although they are of a plausible magnitude; Pearson et al., 2009; Pagani et al., 2011; Foster et al., 2017), and these could be refined to provide a slightly better absolute fit to the data. Orbital variability does not appear to have a major impact on the comparison as shown by the relatively minor impact on the results in the FOAM simulations, and due to the length of the averaged periods of the proxy records.

A major concern identified in the model-data comparison is that even the best simulation pairs for both models do a poor job at recreating the change across the EOT compared to the latitudinal gradient benchmark or even the constant mean benchmark

(Figure 3c). From 55-65 °S, the HadCM3BL and FOAM simulations are mostly in agreement with the mean change observed in the proxy records, however, in the models south of 65 °S there is a strong increase in cooling with poleward latitude, again due to the cooling effect of the ice sheet, with a zonal mean cooling in the range of 10-15 °C at 75 °S. At the Ross Sea site, the S-index proxy suggests only minor cooling of 1 ±5 °C (Passchier et al., 2013). Although, the vegetation records could suggest greater cooling at this site, given the large range in early Oligocene temperature estimates from these records (Francis & Hill, 1996; Raine & Askin, 2001; Passchier et al., 2013, supplementary information), it has not been possible to fully constrain the EOT temperature change with this data.

Critically assessing the proxy records that are included in the compilation could explain some of the differences between the records and the models. For example, it can be unclear as to what area the terrestrial proxies such as the S-index represent, or to what extent this record is affected by reworking. The S-index, like any detrital-based proxy, will suffer to some extent from reworking of older material (Passchier et al., 2013). This residual signal, primarily built up in warmer periods, implies that a warm-bias is likely. Additionally, although the dataset used here was as large as could be compiled at the time of writing, there are still large data gaps spatially and temporally. It is possible the sites around the Ross Sea are part of very localised microclimates, which may not align with the average climate of the large areas covered by a model grid cell.

A second option that could partly explain the model-data discrepancy is that local-regional scale warming signals in response to Antarctic glaciation due to enhanced circulation, deep water formation and sea ice feedbacks (as identified in models by Goldner et al., 2014; Knorr & Lohmann, 2014; Kennedy et al., 2015; Kennedy-Asser et al., 2019; and some of the FOAM simulations used here from Ladant et al. 2014; figure not shown) could be compensating for some of the cooling. When this warming is combined with a $pCO_2$ decline, the models do suggest that some very localised areas (i.e. < 5 grid cells) show little cooling or even warming, while other regions around the world cool more (figure not shown). It is therefore possible that the models are producing a qualitatively realistic result (i.e. relatively less cooling south of 60 °S compared to north of 60 °S), however they do not get the location or magnitude to match the proxy datasets. A potential issue with this hypothesis is that the modelled warming with glaciation was shown in Kennedy-Asser et al. (2019) to be largely reduced with increasing spin-up, suggesting a similar effect could negate the at least some of the warming found in the other models (with the spin-up lengths of Goldner et al., 2014, Knorr & Lohmann, 2014 and Ladant et al., 2014 all ranging between 2,000-3,500 years).

Another significant model-data discrepancy is the strong cooling indicated by $U_{K'37}$ at the Falklands Plateau (Liu et al., 2009; Plancq et al., 2014), which is significantly greater than any cooling recorded at any other Southern Ocean site or at any other site in the Atlantic more broadly (Liu et al., 2009). Although the most recent $TEX_{86H}$ reconstructions suggest more moderate cooling, this could be biased towards summer temperatures (Houben et al., 2019), so the $U_{K'37}$ record cannot be disregarded. This major cooling is hard to explain by any large-scale oceanic process that would be present in these low-resolution models. Even if there were to be a shift in the Antarctic Circumpolar Current and the Antarctic convergence, resulting in cold Southern

Ocean waters reaching the site, surface waters 8 °C cooler lie more than 15 ° further south. As a result, the model simulations presented here would suggest that the major cooling that occurred at this site (assuming it is not due to some other error or bias in the record processing) is due to a geographically restricted (small scale) feature, such as becoming influenced by an upwelling of cold deep and/or intermediate water. Such features are below the resolution of these models and unfortunately cannot be expected to be captured.

A final important consideration is that the temporal averaging of the dataset carried out here could be inappropriate. A number of studies have suggested there was cooling in the several million years prior to the EOT, particularly at high latitudes (e.g. Raine & Askin, 2001; Petersen & Schrag, 2015; Passchier et al., 2016; Carter et al., 2017; Pound & Salzmann, 2017). Even in the high Northern Hemisphere changes have been identified occurring prior to the EOT (e.g. Coxall et al., 2018). These changes could all have a range of different forcings; however, it is possible that some of them are related. Even a global forcing such as atmospheric $p$CO$_2$ decline would potentially have a signal that is detected first at higher latitudes. If there is polar amplification of the cooling signal and if there is a threshold of magnitude at which temperature changes could be identified in the proxy record (or other elements of the Earth system start to respond to the temperature change; e.g. changes in vegetation, weathering or precipitation), then even with a gradual decline in $p$CO$_2$ there could appear to be temporal heterogeneity in the response.

Regardless of whether late Eocene cooling was earlier or simply amplified at higher latitudes, in both cases it is likely that the Ross Sea site experienced significant cooling prior to the EOT. This would support evidence of some tundra vegetation in the region recorded prior to the EOT (Raine & Askin, 2001). It therefore might be necessary to include older records and further split the dataset into additional time slices to capture the climate of Antarctica before any cooling occurred. The only record of this age included in the current dataset is the McMurdo erratic, which suggested temperatures of less than 13 °C (Francis et al., 2009); however, the original location of this fossilised section is unknown and it could represent an area further south or at higher altitude, and thus introduce a cold temperature bias and is not suitable to use in isolation.

Currently, the data compilation is not big enough to allow for such analysis to be carried out; however, this could potentially offer a more appropriate comparison with the equilibrium climate model simulations used here, which are broadly 'warm and ice-free' or 'cool and glaciated'. If this hypothesis is correct, if more comparable records were included for the period pre-cooling and glaciation (e.g. dating from 40 Ma), it is possible that the high latitude change from the mid-late Eocene through to the Oligocene would be greater than that which is shown in Figure 3, closer in line with the model simulations.

# 5 Conclusions

An extensive review of temperature proxy records for the high latitude Southern Hemisphere region before, after and across the EOT was presented and used to evaluate model simulations of the EOT. These simulations came from two different GCMs with different sets of boundary conditions. The best simulations were able to capture spatial patterning of absolute temperature recorded in the late Eocene and early Oligocene proxy datasets. The performances were not as good for the dataset of relative changes across the EOT, due to the models inadequately capturing changes in the latitudinal gradient shown by the data. The latitudinal gradient discrepancy is possibly related to the paucity of data in certain regions (particularly at very high latitude), the time averaging of the proxy records into time slices (with some of the higher latitude changes possibly occurring prior to the EOT), localised climatic effects (e.g. ocean upwelling or ice free coastal microclimates) or possibly because the glaciation of Antarctica results in some localised warming through changes in atmospheric or oceanic circulation that approximately balances the general cooling across the EOT (e.g. due to $p$CO$_2$ decline). If the latter in the case, it would qualitatively support the responses found by HadCM3BL and FOAM, as well as by other models (Goldner et al., 2014; Knorr & Lohmann, 2014). If this is correct, the poorer results in the model-data comparison carried out here may be because the models are simply mis-identifying the areas where the warming occurs.

The best pairs of simulations for modelling the absolute temperatures and relative changes were found by assessing the individual simulations' performance across all time slices for various metrics. This suggests that the best simulations for representing the EOT were by HadCM3BL with an open Drake Passage, AIS expansion and possibly a drop in atmospheric $p$CO$_2$ levels. The poorer fit with the data for the late Eocene and early Oligocene when the Drake Passage is closed suggests the gateway was open for the duration of this period, while an opening of the Drake Passage across the EOT also produces a poor fit with the various datasets compiled here. This suggests the Drake Passage was open prior to the late Eocene and EOT and so opening of the Drake Passage was an unlikely driver of the EOT (in agreement with the results of DeConto & Pollard, 2003; Goldner et al., 2014 etc.).

The performance of FOAM for the early Oligocene time slice was generally better with smaller ice sheet configurations over Antarctica, potentially in agreement with proxy records of ice volume and extent (e.g. Bohaty et al., 2012; Huang et al., 2014; Galeotti et al., 2016). A similar finding is also seen in the HadCM3BL simulations using the Getech palaeogeographies (not shown; Kennedy-Asser et al., 2019); however, as these simulations could be affected by a lack of spin-up, they were not included in the analysis. Further spinning-up those HadCM3BL simulations with multiple ice sheet sizes could provide some interesting insights into whether this climatic fingerprint of a smaller AIS is robust.

These results point towards some interesting conclusions about how the Earth system changed across the EOT, however this work remains a first step upon which further research should be built. An important consideration in interpreting this model-

data comparison is the relative paucity of data available for the region during the EOT (only 14 sites), in combination with records generally showing heterogeneous temperature patterns. Particularly for the normalised RMSE, an important measure for determining if the model is showing the correct spatial patterns, there are only a handful of sites which can be used across all sectors of the Southern Ocean. With the relatively limited data coverage available here, it is possible that these latitudinal

profiles could be biased by anomalous values. However, as noted in Section 2.4, even with the most extreme points omitted for the calculation of the latitudinal gradients for each time slice, no gradient fundamentally changed. Expanding the datasets in the future as more data points become available is a more appropriate method for testing if points used here are anomalous and if the latitudinal profiles are robust.

Future research by the palaeoclimate community will inevitably produce new records in new locations, potentially refining or even correcting older, spurious results or having an impact on the inferred spatial patterns shown in the proxy record. Future work on this research could improve the consistency of the data used, for example in terms of using the same proxy calibrations, age models and definitions of uncertainty, as well as fully accounting for uncertainty in seasonal biases and orbital variations, but that is currently beyond the scope of this paper. To this end, the datasets used here have been uploaded to the Open Science

Framework (Kennedy-Asser, 2019) to aid in the continuation of this research and the expansion of this analysis in the future.

Additionally, future work can also expand upon this analysis by including more model simulations and trialling other metrics and scoring techniques, as palaeoclimate modelling results are often model dependent (Lunt et al., 2012). It is also important to note that the models used here are of relatively low spatial resolution, meaning the spatial averaging of temperature is taken

over a very large area and potential smaller scale ocean changes resulting from changes in ocean gateways may be poorly represented. Therefore, although these simulations are likely to capture large scale climate phenomena, clearly much could be learnt in future research from using higher resolution models.

The challenge in synthesising the many changes that occurred in this large and heterogeneous region across the EOT is huge,

but this research shows that with increased modelling and proxy data results, some convergence of ideas within the palaeoclimate community appears possible.

**Acknowledgements**

ATK-A was funded by NERC (NE/L002434/1). HadCM3BL climate simulations were carried out using the computational facilities of the Advanced Computing Research Centre, University of Bristol (http://www.bris.ac.uk/acrc; Bluecrystal). JBL thanks the CEA/CCRT for providing access to the HPC resources of TGCC (GENCI, allocation 2014-012212). We thank H. K. Coxall, O. Andrews the two anonymous reviewers for their helpful comments in developing the manuscript.

**Data and Code availability**

All of the model data presented in this research along with Matlab scripts used to carry out the analysis are available via the Open Science Framework (Kennedy-Asser, 2019). Further HadCM3BL variables from these simulations (which were not used in this research) are freely available through the University of Bristol's BRIDGE server

5 (https://www.paleo.bristol.ac.uk/ummodel/scripts/papers/Kennedy-Asser_et_al_2019.html). Further variables from and information regarding the FOAM simulations are available from jbladant@gmail.com.

**Author contribution**

ATK-A carried out the analysis and compiled the proxy datasets. DJL helped in the experimental design. PJV and JBL provided

10 the climate model data. JF and VL provided guidance in the interpretation and compilation of proxy data. ATK-A wrote the manuscript with contributions from all authors.

**Competing interests**

The authors declare that they have no conflict of interest.

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
