# Peer review of "Changes in the high latitude Southern Hemisphere through the Eocene-Oligocene Transition: a model-data comparison"

_Climate of the Past, 2019_

## Referee Comment (RC1) · Anonymous Referee #1 · 15 Oct 2019

The aim of the work was to produce a compilation of climatic proxies across the EOT and use this compilation to measure the performance of low-resolution AOGCM of different time slices across the EOT (performed in a previous study by the same lead author). Different boundary conditions are considered, i.e. varying pCO2 and or Antarctic ice sheet extent, opening of closing Drake Passage. The main conclusions of this work is that on one hand, in general the paucity of data cannot really allow for a strong benchmark of AOGCM; on a second hand, the model-data comparison suggest that drop in CO2 have driven the sharp global cooling across EOT rather than the Drake passage opening.

[Figure]

I see this paper as on of the first step necessary for the DEEP MIP project. In fact, proxy validation is critical to measure the ability of models to reproduce different extreme climatic states.

I am not fully convinced by this work because there are critical aspects that are not addressed properly and needs to be clarify and properly justified. At least main points needs to be address in this contribution before elaborating discussion on climatic interpretation. Moreover, those kind of results are also very model dependent:

1- A discussion on the impact of horizontal resolution of the simulations on the modeled variables is necessary before comparing to proxies. How many grid points are considered in the Drake Passage? There is also the Tasman gateway that opens toward 33.5 Ma (Scherer et al., 2015) and also impact on ocean circulation as much as the Drake passage opening. This is not considered here. So it is difficult to conclude that those ocean gateways do not matter too much.

2- Please also discuss the performance of both model for present-day day and other paleo periods? Do they present systematic cold bias as shown here?

3- Where are localized the main biases compared to the proxies? Pacific sector, Atlantic sector? Indian sector?

4- the number of proxies is really little, because only few sites allow to reach those time periods int he Southern Ocean and around the Antarctic margins. However, the number of sites considered in each latitudinal average points, especially in Figure 3, should be indicated to understand the weight of those proxies.

5- Orbital configuration impact is briefly discussed for the model simulations. But what about the proxies? Have they record more glacial transitions or interglacial transitions?

I think that without at least discussing those points in the paper, the comparison made here remains too weak and highly qualitative.

I report only few comments below that are mostly redundant with those general comments.

Major comments:

Page 10 - line 33: Actually the steeper Oligocene gradient discussed here is only caused by one point as shown in Figure 3b. This proxy-based average also has a HUGE uncertainty. Thus I question the validity of this steeper Early Oligocene gradient because only based on one averaged point. See next comment about figure 3.

Figure 3: To really understand the strength of the proxy-based latitudinal T. reconstructions, it would be critical to report below each points shown for Eocene and Oligocene, how many sites are considered in those averaged numbers and associated uncertainty, given their already small numbers. I would be more clear and immediate to the reader than going through all Tables shown in the Supplementary.

Resolution of simulations: In general the impact of coarse resolution simulation is not discussed. However, some recent on-going works (e.g. from Isabel Sauermilch) carried out with high-resolution ocean simulations definitely suggest different conclusions than you work (ISAES or ICP meetings 2019 presentations). This point highlight the fact that the interpretation of EOT in models is highly model dependent and resolution dependent. A discussion on this aspect here is necessary before trying to interpret the difference between your model and the data.

Minor comments:

Figure 1: please insert intermediate tick mark on Y axis to help the reader understand the range of temperature for each sites.

Table 1: Provide the nominal averaged horizontal resolution at high latitudes for both models in order to make it more understandable. Also because horizontal resolution impacts a lot on those reconstructions. . ... Please make it more clear here in this table.

Figure 4: in the caption it is indicated that simulations considered in this figure are "ice free", however, I see that for the Oligocene, simulations are reported with some

antarctic ice sheet. Please correct the caption or clarify.

---

## Referee Comment (RC2) · Anonymous Referee #2 · 29 Nov 2019

**Review of the paper entitled "Changes in the high latitude Southern Hemisphere through the Eocene-Oligocene Transition: a model-data comparison "by Alan T. Kennedy-Asser et al.**

I.    General comments:

There has been a long lasting controversy to assess the major driver of the EOT occurring 34 Ma ago between the opening of the Drake straight and a threshold on the CO2 decrease.

This paper aims to cope with the interesting issue of the comparison of complex coupled low resolution AOGCMs to a multi proxy data base for the period spanning from before the EOT (LE: late Eocene) to after this major event (EO: Early Oligocene).

It is an important issue because there is a gap between the world of data climatic reconstruction including error bars on dating and intensity of temperature changes and the world of the models which are equilibrium climate with prescribed conditions. Therefore building bridges between these two worlds is not an easy task.

This paper is a contribution to bring new constraints in this debate. The authors compiled various annual reconstructions derived from many different proxies over the High Latitudes for 3 periods spanning over several million years (14 sites: 4 terrestrial and 10 oceanic). On the other hand they establish metrics and benchmarking to validate two series of simulations run by low resolution AOGCMs. This is a first attempt to conduct such an approach. As such, it is a pioneer work which indeed depicts strong points and severe limitations.

The major results of this manuscript are

1 to establish the regional pattern of temperatures for high latitudes of southern hemisphere for the 3 key periods: late Eocene, Early Oligocene and during the transition.

2 to diagnose through a series of sensitivity simulations testing different boundary conditions, which are the most appropriate to fit the data. To achieve this goal the authors establish different matrices and benchmarking.

Despite showing new consistent results, there are some caveats in the manuscript.

1 concerning the discussion of the database

2 concerning the model results/data comparison and interpretations which are sometime confusing and unclear

In my detailed review, I will describe these strong and weak points. My general comment is that this manuscript is worth for publication because it is an interesting study, well documented with many supplementary information, but in is current state there is room for improvement. I suggest major modifications and clarifications that are described below.

II.    Detailed review:

**Abstract**

The authors should specify that the database includes 14 sites (4 terrestrial and 10 oceanic) and that two low resolution models are used.

The sentence on the comparison of zonal mean temperature is a bit confusing. As far as I understand, the models are not reproducing the behavior depicted by the data during EOT. Moreover they underestimate the temperature over high latitude for both LE and EO.

The second part of the abstract is consistent with the results depicted in the manuscript.

The major results are not really summarized in the sentence concerning zonal temperature changes.

My reading is that there are clear underestimations of model results for both periods in terms of absolute value but a better agreement when using the relative metric.

The change of latitudinal slope of temperatures pinpointed by the data is not reproduced by both models

**1. Introduction**

The reference to Lunt et al., 216 is perfectly appropriate, but, as the statement is very general, other references could also be included.

During broad time slices as LE (36.4-34 Ma) and EO (33.2-32 Ma), there are many orbital cycles that produce by themselves a large uncertainty (see PLIOMIP1) when comparing to data. Especially after the EOT when AIS (Antarctic Ice Sheet) is also modulated by astronomical forcing factors. This point has to be better described and discussed.

Moreover the series of simulation have been conducted differently especially concerning the cryosphere, which should be discussed in more detail.

**2. Method**
**2.1. Data synthesis**

**2.1 data synthesis**

Concerning the establishment of the specific database, used for benchmarking, of annual temperature at high latitude of Southern Hemisphere for LE and ETO, these reconstructions are derived from many different proxies and therefore

1. Not sure that these different proxies measure the annual temperature. Because temperatures are reconstructed using various proxies, certainly calibration issues should be discussed.
2. These reconstructions include a contribution of the high frequency (astronomical frequencies at the scale of tens of kyr). But this part is not accounted for in the models which are run with prescribed orbital forcing factors.
3. A good illustration of the difficulty of such a comparison is the time to establish a reliable multiproxy data base useful for LGM as MARGO. This effort of a large community took some time due to the necessity to calibrate the different proxies and intercompare them in a consistent chronological framework. Moreover, similarly to PLIOMIP1, the data/model comparison spanning for this stage over 300ky, suffers from variability due to changes of orbital forcing factors which inhibits accurate model/data comparison.

Therefore, the authors should discuss more clearly these limitations shown in fig1 and discuss how to improve this first attempt.

**2.2 Model simulations**

The interest of the manuscript is to use a series of sensitivity experiments on the different boundary conditions (ice sheet, Drake opening/closing, CO2). Indeed CO2 and AIS extent are not independent and the impact of each factor is maybe not cumulative, these 2 aspects should be discussed.

**2.3 Metrix**

 Why to establish these absolute and normalized metrics? The authors should clarify this question.

For warm periods, models are generally unable to reproduce the zonal temperature gradient and very frequently, models largely underestimate the temperature reconstructions based on different proxy data at high latitudes.

This is also illustrated in this paper for both periods, therefore another normalized metric which is appropriate to avoid this problem is used in this manuscript which is fine for me but this should be clarified.

Whatever the metric used, the spatial average for each data location over 9-grid box represents a large region around 1000x1000km2 which is anyway huge and should be discussed.

**2.5 Benchmarking**

Introducing different matrices and developing a benchmarking is appropriate to provide a quantified model-data comparison. What is the benchmarking of both models for present day simulations? It would be interesting to know if some biases may explain the response of both models for deep time simulations.

**3 Results**

Fig3 and section 3.1   remain unclear. I don't understand whether there is a real robust quantification of the "qualitative agreement" invoked by the authors.

**Late Eocene and Early Oligocene.**

These sections and Fig4 summarize well the series of model simulations and the comparison with the database through different metrics and benchmarking. Nevertheless, cold biases due to the inappropriate capability of the models to reproduce warm conditions during warm climate are clearly illustrated using absolute metric, but not discussed in terms of consequences. For instance, the large extent of the AIS contributes to amplify the initial cold bias.  Therefore, the ice sheet development is not favored in EOT context. Similarly, open Drake is also favored to compensate the original cooling bias. The authors use normalized metric to increase the score and diagnose the best boundary conditions.
The influence of the Drake straight opening, the pCO2 drawdown and the sensitivity to the extension of the ice sheet are expected. More discussion on the fact that the sensitivity of both models to opening of Drake and to the AIS reconstructions should be given, especially the AIS asynchronously

computed in Foam is smaller for early Oligocene. This has for consequence to produce less cooling and due to a cold bias it fits better with data. Therefore, the results have to be discussed not only in terms of benchmarking but more generally accounting for specific bias of each model.

**EOT**

The transition from late Eocene to early Oligocene is particularly interesting to analyze through all pairs of sensitivity experiments, either to diagnose better scenarios or to disregard some of them. In this section, because we deal with climate variations, the standard metrix has not the same drawbacks than previously. An interesting point of this contrasted section if to show that the poorest correlation is obtained for the Drake passage opening.

**4 Discussion**

This transition seems driven by many factors. Is it reasonable to modify the pCO2 to get a better fit to data? There is still large room for improvement to correctly simulate reasonable zonal thermal gradient for Late Eocene and early Oligocene which are necessary conditions to understand EOT. Moreover more information quantitatively depicting how different are the equilibriums for both model simulations is needed. What are the criteria used for the spin up phase? Which consequences it may have on the results provided in this study?

**5 Conclusion**

The build-up of this data set of high latitude southern hemisphere annual temperatures is a first attempt to validate model results of the EOT. Nevertheless these 14 points correspond to reconstructions with different proxies. Moreover, coarse resolution models are used, the spatial averaging includes 9 grid boxes which correspond to a square of around 1000 km2 side. Both features limit the model data comparison.

Nevertheless, these results bring some credit to the respective contributions of AIS development, CO2 decrease and Drake opening to the EOT and conclude to a minor role played by this latter. This conclusion is therefore interesting but remains not completely convincing due to many uncertainties that the authors describe.

The constraints on the development of the AIS when coupling climate and cryosphere during the EOT are also a promising way to capture the evolution of a whole system in which the AIS is active.

Increasing model resolution, coupling GCM and ice sheet models, increasing the number of data through EOT, better accounting for annual temperature variability induced by astronomical forcing factors, all these features are important ways to continue this pioneering study.

**Figures**

On fig1, for sites that benefited from several temperature reconstructions as Ross or Falklands Plateau, the spread is large which shows that a future work of intercalibration and synthesis like MARGO remains necessary. From fig. 1, it is clear that there is a tendency to lower temperatures for Oligocene, however, the values for both periods are compatible within the error bars. Indeed in fig.2, most of the values for the temperature differences are compatible with zero. However, there is a clearly negative value for $U^K_{37}$, but it is not clear why the error bar for this value is so small.

figure 3: caution it is not North but South on the axis title.

Despite the large uncertainty of the models, the models tend to underestimate the high latitude temperatures (fig. 3 a and b), and the effect is larger for Oligocene. Hence, the models predict in overall a larger cooling at high latitude than the data (fig 3c). The surprising behavior depicted in the text (larger cooling over mid latitude than for high latitudes) is indeed not reproduced by the model. However, the signal for such a latitudinal trend seems very weak and considering the model uncertainty and the error bars on the data, the model results seem to be compatible with each data point.

---

## Author Comment (AC1) · 23 Dec 2019

We would like to thank the reviewer for taking the time to review this paper. Five main points were identified by the reviewer:

1) Regarding the horizontal resolution of the models and how they resolve the gateways 2) How the models perform for the present day or other palaeo periods 3) If regions have specific model-proxy data anomalies 4) The paucity of data in the dataset 5) The potential impact of orbital configurations

These will be addressed in turn below:

[Figure]

1) The model resolution indeed could impact how the models' respond to changes in ocean gateways. We can add further discussion of the results from other studies using higher resolution, e.g. eddy resolving ocean models to highlight this. Due to time and computational constraints, we will not be able to carry out high resolution simulations for the current paper, but hope that our clear methods and openly available datasets could be used by future researchers with higher resolution model results to verify our findings. Additionally, as we noted in Section 4.1., specifically focussing on changes to the Tasman Seaway would be interesting for further research.

2) Both of these models have been verified against present day and other palaeo time periods and we can add discussion and references about their performance. Both capture major characteristics of the present day climate and reasonable variability, but notably both have cold biases over the high latitude northern hemisphere in the winter. This could account for why high latitude temperature biases exist in these models for palaeo time periods.

3) There is no specific sector (e.g. Pacific, Atlantic, Indian) that is particularly poor for all time slices. For the late Eocene and early Oligocene, the models are too cold at all sites. For the change across the EOT, sometimes models underestimate and sometimes overestimate the change, so this is harder to generalise. The New Zealand site is particularly poor for the late Eocene and early Oligocene. For the change across the EOT, the Maud Rise record has the largest anomaly between the model simulations and the data.

4) We have been able to recalculate the latitudinal temperature gradients systematically omitting single points from the datasets to test if they are particularly biased by single extreme or anomalous points. This would suggest that there is no significant difference between the gradients in the late Eocene and early Oligocene, with the early Oligocene data being more affected by extreme values. However, this analysis does not fundamentally change any of the conclusions of the paper. This is the best solution to this issue we can provide without generating many new, independent proxy records.

5) We have been able to redo our analysis using FOAM simulations with different orbital configurations and found that this has a relatively minor impact on the overall fit to the data (less than other factors such as the ice sheet size or pCO2 level). While we cannot say exactly by what magnitude the proxy records might be affected by orbital variability, it could be assumed that the magnitude of the effect would be similar to that in the model. We expect the effect on many of the proxies to be relatively small as they are generally time averaged over periods longer than single orbital cycles.

The other minor comments (mainly relating to figure presentation) can be easily addressed.

---

## Author Comment (AC2) · 23 Dec 2019

We would like to thank the reviewer for taking the time to provide this detailed review of this paper. Similar key points were brought up to reviewer 1, including:

1) Uncertainty in the datasets, particularly relating to orbital variability and potential seasonal biases 2) The low resolution of the model simulations 3) How the models compare to the present day and how this might affect the results

Furthermore, they also ask for further details on aspects of the model setup and spin-up, as well as further discussion of results.  Firstly, we address the main points that

overlap with reviewer 1 below:

1) To understand this, we have repeated our analysis using model simulations with varying orbits and also using only summer temperatures. Although we cannot definitively show the impact of orbital variation on individual records without a comprehensive reinvestigation of the records (which would be beyond the scope of the paper), the additional modelling results show that the impact of variations in the orbit are relatively minor and do not affect the core conclusions of the paper. Likewise for summer temperatures, although this would correct some of the absolute temperature bias for the late Eocene and early Oligocene, it does not make a major difference in improving the models' fit with the data or siginifcantly change the findings. Indeed, the summer model data fits other aspects of the datasets much poorer. There is considerable ongoing debate about the potential impact of seasonality in proxy records (e.g. Hollis et al., 2019, GMD) and so for this paper we would simply start with the assumption that the difference would be comparable to the proxy uncertainty that is already included. We acknowledge that future work could expand and further quantify the proxy data uncertainty (hence why we made our datasets available for other researchers to use and build upon), however a thorough review would be beyond the scope of this paper.

2) As with the comments for reviewer 1, we can add discussion relating to how higher resolution modelling could affect the results. Specifically to answer the comments of reviewer 2, we can add further discussion of how we only expect these models to pick up very large scale climate variability.

3) As we noted in the response for reviewer 1, although both of these models capture many aspects of the modern day climate well, they do show a cold temperature bias in the high latitude Northern Hemisphere during winter. The causes of this are not known but likely contribute to the cold biases found in the model-data comparison carried out here. Understanding why models show these cold biases remains the focus of ongoing work (e.g. DeepMIP). References and discussion about this can be easily added to the text.

We can easily add further discussion of the model setup and spin-up so that readers do not have to look up the respective references. It is worth clarifying that in these simulations the ice sheets are non-interactive and cannot melt or expand. Further discussion of the results and why certain forcings are particularly poor can be added. We expect that both sets of model simulations are adequately spun-up and further spin-up would not change the core conclusions of the paper.

Other comments relating to figures and emphasis in the discussion can easily be corrected.

---

## Author Response (AR1)

Title: Changes in the high latitude Southern Hemisphere through the Eocene-Oligocene Transition: a model-data comparison
Author(s): Alan T. Kennedy-Asser et al.
MS No.: cp-2019-112

Dear Dr Donnadieu,

Please find attached our response to the two reviewers' comments as well as a tracked-changes version of the paper.

We have made considerable expansions of the analysis particularly with regards to understanding orbital and seasonal variability in the models and how this might affect the model-data comparison. We also did further work to assess if the latitudinal gradients found in the dataset were robust or sensitive to the relative paucity of data (which we had previously recommended while discussing 'future work' in the original version of the paper). Doing this sensitivity analysis highlighted that (in agreement with the reviewers' concerns) the Oligocene time slice latitudinal temperature gradient was particularly influenced by one extreme and uncertain point, and so the difference between the absolute Eocene and Oligocene gradients was insignificant. This does not fundamentally change the results of the paper, but has been very worthwhile to carry out from our point of view and we have reweighted our discussion of this accordingly (Section 3.1).

We have highlighted particular biases in the models for the modern day from previous model description papers which are relevant for interpreting the results of this paper (such as cold high latitude biases over the Northern Hemisphere; Section 2.2). However, we also feel that the objective of this paper is not an extensive model validation compared to modern day observations and so have not added any new analysis in this regard.

Similarly, we have tried to further acknowledge that these models are very coarse resolution and that this could be affecting their realism in representing flow through Southern Ocean gateways or their response to gateway changes. However, without including whole new sets of model simulations which we do not currently have access to, we cannot explicitly quantify these differences. The purpose of providing the proxy datasets exactly as were used here as an open source supplement was that other modelling groups who are using higher resolution models could easily recreate the evaluation we have carried out to show if the conclusions are sensitive to the model resolution.

I hope you find these changes justifiable and satisfactory. I look forward to hearing your and the reviewers' responses.

Sincerely,
Alan

Reviewers' comments are in black
Authors' responses are in green
*"Changes to text are in italics and quotation marks"*

**Review 1:**

The aim of the work was to produce a compilation of climatic proxies across the EOT and use this compilation to measure the performance of low-resolution AOGCM of different time slices across the EOT (performed in a previous study by the same lead author). Different boundary conditions are considered, i.e. varying pCO2 and or Antarctic ice sheet extent, opening of closing Drake Passage. The main conclusions of this work is that on one hand, in general the paucity of data cannot really allow for a strong benchmark of AOGCM; on a second hand, the model-data comparison suggest that drop in CO2 have driven the sharp global cooling across EOT rather than the Drake passage opening.

I see this paper as on of the first step necessary for the DEEP MIP project. In fact, proxy validation is critical to measure the ability of models to reproduce different extreme climatic states.

I am not fully convinced by this work because there are critical aspects that are not addressed properly and needs to be clarify and properly justified. At least main points needs to be address in this contribution before elaborating discussion on climatic interpretation. Moreover, those kind of results are also very model dependent:

1) A discussion on the impact of horizontal resolution of the simulations on the modelled variables is necessary before comparing to proxies. How many grid points are considered in the Drake Passage? There is also the Tasman gateway that opens toward 33.5 Ma (Scherer et al., 2015) and also impact on ocean circulation as much as the Drake passage opening. This is not considered here. So it is difficult to conclude that those ocean gateways do not matter too much.

We have added further discussion about the coarse horizontal resolution of the models in Section 4.1, as well as noting the width of the Drake Passage for each model in Table 1.

*"It is important to bear in mind that this result was obtained from a relatively low-resolution model. With higher resolution models, it is possible that changes in modelled ocean circulation and atmospheric response could be very different, particularly given that much smaller changes in the Southern Ocean gateways than were modelled here could have occurred across the EOT (e.g. Viebahn et al., 2016) For this paper, it was not feasible to use higher resolution models for such a range of boundary conditions and length of simulations, and this should remain an important priority in future research."*

Potential changes in the Tasman Seaway is a good point and we have expanded the acknowledgement opening in Section 4.1. of the discussion that this could have a different response to Drake Passage:

*"However, it should be noted that proxy evidence and reconstructions suggest the Tasman Seaway deepened close to, but probably prior to the EOT (e.g. Stickley et al., 2004; Scher et al., 2015,*

*Houben et al., 2019) and this could have different implications on the climate (which are potentially more consistent with temperature proxy records compiled here) from the results shown for Drake Passage opening. The preconditioning effects of widening and deepening the Tasman Seaway therefore could be of interest to focus on with future model comparisons."*

2) Please also discuss the performance of both model for present-day day and other paleo periods? Do they present systematic cold bias as shown here?

A summary of previous work benchmarking the models against other time periods including the present day has been added in section 2.2:

*"For the present day, HadCM3BL is shown to perform comparably to CMIP5 models in terms of a number of global mean variables, although it produces a moderate cold bias globally, with high (northern) latitudes being too cold because of an exaggerated seasonal cycle and overly cold winter (Valdes et al., 2017). This bias is similar to other higher resolution variants of the model (Valdes et al., 2017). FOAM has been shown to capture most of major characteristics of present-day climatology (Jacob, 1997; Liu et al., 2003) as well as reasonable climate variability (Wu & Liu, 2005). As HadCM3BL, FOAM exhibits a cold high latitude bias in the Northern Hemisphere, in particular in winter (Gallimore et al., 2005)."*

3) Where are localized the main biases compared to the proxies? Pacific sector, Atlantic sector? Indian sector?

There is no specific sector that is particularly poor. The New Zealand site is consistently poor, but other sites/records also show large errors for certain time slices. Discussion of this has been added through out Section 3.

Section 3.1:
*"This is consistent across all sites, although the fit with the New Zealand records is particularly poor"*

Section 3.2:
*"The New Zealand records are poorly represented again by all models, but there are also issues representing the Maud Rise, East Tasman Plateau and one of the Ross Sea records."*

Section 3.3:
*"For this dataset, there is not a clear picture of modelled changes being over- or underestimated relative to the proxy records. The largest error for all models is in representing the large cooling shown at the Falklands Plateau."*

4) the number of proxies is really little, because only few sites allow to reach those time periods int he Southern Ocean and around the Antarctic margins. However, the number of sites considered in each latitudinal average points, especially in Figure 3, should be indicated to understand the weight of those proxies.

Throughout the text, we have increased the acknowledgement of the limited number of proxies (10 ocean sites; 4 terrestrial sites). All of the sites and proxies shown in Figures 1 and 2 are included in Figure 3: values of the number of data points, *n*, have been added to Figure 3.

5) Orbital configuration impact is briefly discussed for the model simulations. But what about the proxies? Have they record more glacial transitions or interglacial transitions? I think that without at least discussing those points in the paper, the comparison made here remains too weak and highly qualitative.

We have greatly expanded our discussion of the impact of orbits in the models by using FOAM simulations with both warm and cold summer orbits. While we cannot say exactly by what magnitude the proxy records might be affected by orbital variability, we have included some possible assumptions that the magnitude of the effect could be similar to that in the model and that we expect the effect on many of the proxies to be relatively small as they are generally time averaged over periods longer than single orbital cycles. See Section 1 (paragraph beginning "Equilibrium climate simulations also simplify orbital variations…") and Section 2.1 (paragraph beginning "It is likely that some seasonal (summer) bias…").

I report only few comments below that are mostly redundant with those general comments.

Major comments:

Page 10 - line 33: Actually the steeper Oligocene gradient discussed here is only caused by one point as shown in Figure 3b. This proxy-based average also has a HUGE uncertainty. Thus I question the validity of this steeper Early Oligocene gradient because only based on one averaged point. See next comment about figure 3.

This is a good point. We recalculated the latitudinal gradient for each time slice systematically removing single data points (as we had previously recommended doing in future work). Doing this gives certain situations where the Oligocene gradient is both shallower and steeper than the Eocene gradient. Given the difference between the gradients of these time slices is insignificant, we have removed the emphasis from this point in the discussion in Section 3.1:

*"In addition to the latitudinal gradient calculated for the full dataset for each time slice, an uncertainty range for the gradient was calculated by systematically omitting single points from the comparison. The absolute temperature profiles in the late Eocene and early Oligocene proxy datasets show colder temperatures at higher latitudes than mid-latitudes, as would be expected. The latitudinal gradient is comparable between the early Oligocene (0.54 °C °N-1; range 0.30 to 0.63 °C °N-1) and the late Eocene (0.49 °C °N-1; range 0.45 to 0.54 °C °N-1), with the Oligocene gradient more uncertain due to the greater variability in the proxy records."*

Figure 3: To really understand the strength of the proxy-based latitudinal T. reconstructions, it would be critical to report below each points shown for Eocene and Oligocene, how many sites are considered in those averaged numbers and associated uncertainty, given their already small numbers. I would be more clear and immediate to the reader than going through all Tables shown in the Supplementary.

For clarity, the number of values included in each time slice have been added, along with the latitudinal gradient.

Resolution of simulations: In general the impact of coarse resolution simulation is not discussed. However, some recent on-going works (e.g. from Isabel Sauermilch) carried out with high-resolution ocean simulations definitely suggest different conclusions than you work (ISAES or ICP meetings 2019 presentations). This point highlight the fact that the interpretation of EOT in models is highly model dependent and resolution dependent. A discussion on this aspect here is necessary before trying to interpret the difference between your model and the data.

As noted for the previous point, we have added further discussion about the coarse horizontal resolution of the models in sections 2.3 and 5 and acknowledged that these results will likely be model dependent in section 5.

Minor comments:

Figure 1: please insert intermediate tick mark on Y axis to help the reader understand the range of temperature for each sites.

Done.

Table 1: Provide the nominal averaged horizontal resolution at high latitudes for both models in order to make it more understandable. Also because horizontal resolution impacts a lot on those reconstructions. Please make it more clear here in this table.

This has been added along with the width of the Drake Passage for both models to Table 1.

Figure 4: in the caption it is indicated that simulations considered in this figure are "ice free", however, I see that for the Oligocene, simulations are reported with some antarctic ice sheet. Please correct the caption or clarify.

The labelling of Figures 4 and 5 have been updated to improve clarity.

**Review 2:**

I. General comments:

There has been a long lasting controversy to assess the major driver of the EOT occurring 34 Ma ago between the opening of the Drake straight and a threshold on the CO2 decrease. This paper aims to cope with the interesting issue of the comparison of complex coupled low resolution AOGCMs to a multi proxy data base for the period spanning from before the EOT (LE: late Eocene) to after this major event (EO: Early Oligocene).

It is an important issue because there is a gap between the world of data climatic reconstruction including error bars on dating and intensity of temperature changes and the world of the models which are equilibrium climate with prescribed conditions. Therefore building bridges between these two worlds is not an easy task.

This paper is a contribution to bring new constraints in this debate. The authors compiled various annual reconstructions derived from many different proxies over the High Latitudes for 3 periods spanning over several million years (14 sites: 4 terrestrial and 10 oceanic). On the other hand they establish metrics and benchmarking to validate two series of simulations run by low resolution AOGCMs. This is a first attempt to conduct such an approach. As such, it is a pioneer work which indeed depicts strong points and severe limitations.

The major results of this manuscript are
  1) to establish the regional pattern of temperatures for high latitudes of southern hemisphere for the 3 key periods: late Eocene, Early Oligocene and during the transition.
  2) to diagnose through a series of sensitivity simulations testing different boundary conditions, which are the most appropriate to fit the data. To achieve this goal the authors establish different matrices and benchmarking.

Despite showing new consistent results, there are some caveats in the manuscript.
  1) concerning the discussion of the database
  2) concerning the model results/data comparison and interpretations which are sometime confusing and unclear

In my detailed review, I will describe these strong and weak points. My general comment is that this manuscript is worth for publication because it is an interesting study, well documented with many supplementary information, but in is current state there is room for improvement. I suggest major modifications and clarifications that are described below.

II. Detailed review:

Abstract:

The authors should specify that the database includes 14 sites (4 terrestrial and 10 oceanic) and that two low resolution models are used.

This has been added.

The sentence on the comparison of zonal mean temperature is a bit confusing. As far as I understand, the models are not reproducing the behavior depicted by the data during EOT. Moreover they underestimate the temperature over high latitude for both LE and EO. The second part of the abstract is consistent with the results depicted in the manuscript. The major results are not really summarized in the sentence concerning zonal temperature changes. My reading is that there are clear underestimations of model results for both periods in terms of absolute value but a better agreement when using the relative metric. The change of latitudinal slope of temperatures pinpointed by the data is not reproduced by both models

The abstract text has been modified, which hopefully clarifies which aspects of the models' performance are better and worse.

Introduction:

The reference to Lunt et al., 216 is perfectly appropriate, but, as the statement is very general, other references could also be included.

More references have been added.

During broad time slices as LE (36.4-34 Ma) and EO (33.2-32 Ma), there are many orbital cycles that produce by themselves a large uncertainty (see PLIOMIP1) when comparing to data. Especially after the EOT when AIS (Antarctic Ice Sheet) is also modulated by astronomical forcing factors. This point has to be better described and discussed.

A paragraph has been added to the introduction discussing how orbital cycles can be represented in model simulations and how they might also influence proxy records (section 1, paragraph beginning "Equilibrium climate simulations also simplify orbital variations…").

Moreover the series of simulation have been conducted differently especially concerning the cryosphere, which should be discussed in more detail.

A paragraph has been added to section 2.2 discussing how the ice sheets are implemented in the models (paragraph beginning "It should be noted that for both models the Antarctic ice sheets…").

Method:
Data synthesis:

Concerning the establishment of the specific database, used for benchmarking, of annual temperature at high latitude of Southern Hemisphere for LE and ETO, these reconstructions are derived from many different proxies and therefore

1) Not sure that these different proxies measure the annual temperature. Because temperatures are reconstructed using various proxies, certainly calibration issues should be discussed.

2) These reconstructions include a contribution of the high frequency (astronomical frequencies at the scale of tens of kyr). But this part is not accounted for in the models which are run with prescribed orbital forcing factors.

3) A good illustration of the difficulty of such a comparison is the time to establish a reliable multiproxy data base useful for LGM as MARGO. This effort of a large community took some time due to the necessity to calibrate the different proxies and intercompare them in a consistent chronological framework. Moreover, similarly to PLIOMIP1, the data/model comparison spanning for this stage over 300ky, suffers from variability due to changes of orbital forcing factors which inhibits accurate model/data comparison.

Therefore, the authors should discuss more clearly these limitations shown in fig1 and discuss how to improve this first attempt.

1) This has been added to Section 2.1: *"It is likely that some seasonal (summer) bias is incorporated into marine proxy records particularly at high latitudes, where light and temperature may become limiting in certain periods. In contrast, for SAT estimates based on*

*vegetation, other conditions such as high atmospheric pCO2 may actually push the thermal tolerances of plants to levels higher than the present-day training set, potentially leading to (winter temperature) underestimates in reconstructions (e.g. Royer et al., 2002). Indeed, the extent of these biases is debated (Hollis et al., 2019) and may not be greater than the calibration errors that are already incorporated."*

2) To explore this point, we have expanded the analysis to also include FOAM simulations with alternative orbits. Although we cannot definitively show the impact of orbital variation on individual records without a comprehensive reinvestigation of the records (which would be beyond the scope of the paper), the additional modelling results show that the impact of variations in the orbit are relatively minor and do not affect the core conclusions of the paper. We hope that this addresses this point.

3) As above.

Model simulations:

The interest of the manuscript is to use a series of sensitivity experiments on the different boundary conditions (ice sheet, Drake opening/closing, CO2). Indeed CO2 and AIS extent are not independent and the impact of each factor is maybe not cumulative, these 2 aspects should be discussed.

We have added further explanation of the models' setups in section 2.2, reiterating that the AIS is non-interactive in both. The method we use to model the change across the EOT necessitates the growth of some ice over Antarctica, potentially in combination with other forcings. By showing the combined effect of e.g. $p\text{CO}_2$ drop and AIS growth we are acknowledging that these changes in the Earth system (although they can be thought of as 'forcings' in a climate model) are not independent and can act cumulatively.

Metric:

Why to establish these absolute and normalized metrics? The authors should clarify this question. For warm periods, models are generally unable to reproduce the zonal temperature gradient and very frequently, models largely underestimate the temperature reconstructions based on different proxy data at high latitudes. This is also illustrated in this paper for both periods, therefore another normalized metric which is appropriate to avoid this problem is used in this manuscript which is fine for me but this should be clarified.

We have added some clarification that high latitude warm biases are typical in palaeoclimate reconstructions in these models in section 2.3:

*"The purpose of removing the mean is so the model performance is not primarily judged against systematic warm or cold biases, the latter of which are typical at high latitudes in palaeoclimate simulations of past warm climates (Huber & Caballero, 2011; Lunt et al., 2012). This 'normalised' RMSE, defined in Eq. (3), instead evaluates the spatial pattern of temperature in the Southern Ocean."*

Whatever the metric used, the spatial average for each data location over 9-grid box represents a large region around 1000x1000km2 which is anyway huge and should be discussed.

We have added a note on the size of the region which is averaged for the modelled values in section 2.3:

*"Given the relatively coarse resolution of these models, this represents a very large area (ranging 2.25-6.35 $\times$ 10$_5$ km$_2$). This method will therefore only capture large scale climate variability and not local variations."*

Benchmarking:

Introducing different matrices and developing a benchmarking is appropriate to provide a quantified model-data comparison. What is the benchmarking of both models for present day simulations? It would be interesting to know if some biases may explain the response of both models for deep time simulations.

A brief discussion of how these models perform for the present day and other palaeo periods has been noted in section 2.2 with further references provided, however a thorough review of this is beyond the scope of the paper.

*"These models are all relatively low resolution and less complex than some others that have been used in recent studies (e.g. Hutchinson et al., 2018; Baatsen et al., 2018); however, they are still regularly used in palaeoclimate research (e.g. Goddéris et al., 2017; Farnsworth et al., 2019; Saupe et al., 2019). For the present day, HadCM3BL is shown to perform comparably to CMIP5 models in terms of a number of global mean variables, although it produces a moderate cold bias globally, with high (northern) latitudes being too cold because of an exaggerated seasonal cycle and overly cold winter (Valdes et al., 2017). This bias is similar to other higher resolution variants of the model (Valdes et al., 2017). FOAM has been shown to capture most of major characteristics of present-day climatology (Jacob, 1997; Liu et al., 2003) as well as reasonable climate variability (Wu & Liu, 2005). As HadCM3BL, FOAM exhibits a cold high latitude bias in the Northern Hemisphere, in particular in winter (Gallimore et al., 2005)."*

Results:

Fig3 and section 3.1 remain unclear. I don't understand whether there is a real robust quantification of the "qualitative agreement" invoked by the authors.

Agreed that this statement could be unclear and as it was not essential for the narrative of the paper has been removed.

Late Eocene and Early Oligocene:

These sections and Fig4 summarize well the series of model simulations and the comparison with the database through different metrics and benchmarking. Nevertheless, cold biases due to the inappropriate capability of the models to reproduce warm conditions during warm climate are clearly illustrated using absolute metric, but not discussed in terms of consequences. For instance, the large extent of the AIS contributes to amplify the initial cold bias. Therefore, the ice sheet development is not favored in EOT context. Similarly, open Drake is also favored to compensate the original cooling bias. The authors use normalized metric to increase the score and diagnose the best boundary conditions.

The influence of the Drake straight opening, the pCO2 drawdown and the sensitivity to the extension of the ice sheet are expected. More discussion on the fact that the sensitivity of both models to opening of Drake and to the AIS reconstructions should be given, especially the AIS asynchronously computed in Foam is smaller for early Oligocene. This has for consequence to produce less cooling and due to a cold bias it fits better with data. Therefore, the results have to be discussed not only in terms of benchmarking but more generally accounting for specific bias of each model.

We have significantly expanded our results section to include both seasonal (summer) variants for both models and orbital variation in the FOAM simulations, as well as discussing more the general cold biases found in both models. These extra results show that there is not a simple narrative, for example that the model is too cold and therefore a bigger AIS or cold orbit makes the fit worse. In many cases the cold orbit for FOAM actually performs better than the warm orbit, and for summer temperatures in the early Oligocene a large AIS gives a better fit to the data in terms of the normalised RMSE than a small AIS.

We hope that the expanded discussion in Sections 3 and 4 helps illustrate some of these points and address these review concerns.

EOT:

The transition from late Eocene to early Oligocene is particularly interesting to analyze through all pairs of sensitivity experiments, either to diagnose better scenarios or to disregard some of them. In this section, because we deal with climate variations, the standard metrix has not the same drawbacks than previously. An interesting point of this contrasted section if to show that the poorest correlation is obtained for the Drake passage opening.

Discussion:

This transition seems driven by many factors. Is it reasonable to modify the pCO2 to get a better fit to data? There is still large room for improvement to correctly simulate reasonable zonal thermal gradient for Late Eocene and early Oligocene which are necessary conditions to understand EOT.

This is a good point. We have re-weighted the discussion to put more emphasis on reducing the mismatch on latitudinal temperature gradients as opposed to fine tuning $p$CO2.

Moreover more information quantitatively depicting how different are the equilibriums for both model simulations is needed. What are the criteria used for the spin up phase? Which consequences it may have on the results provided in this study?

Discussion of the spin-up has been added to Section 2.2:

*"The model spin-up period also differs between the two studies, with the HadCM3BL simulations being significantly longer. The HadCM3BL simulations were selected from Kennedy-Asser et al. (2019) based upon their extended spin-up, meaning the modelled results are expected to be highly robust with negligible trends to bias the conclusions. FOAM simulations have been integrated for 2000 years and are in equilibrium in the upper ocean. Small cooling trends exist in the deep ocean but the rates of temperature change are smaller than 0.1°C/century, which is a criterion regularly*

*used to define quasi-equilibrium (e.g., Lunt et al. 2017). The conclusions are therefore expected to be robust."*

Conclusion:

The build-up of this data set of high latitude southern hemisphere annual temperatures is a first attempt to validate model results of the EOT. Nevertheless these 14 points correspond to reconstructions with different proxies. Moreover, coarse resolution models are used, the spatial averaging includes 9 grid boxes which correspond to a square of around 1000 km2 side. Both features limit the model data comparison.

*The number of sites has been re-emphasised in the conclusion along with two sentences acknowledging the low resolution of these models.*

*"It is also important to note that both of the models used are of relatively low spatial resolution, meaning the spatial averaging of temperature is taken over a very large area and potential smaller scale ocean changes resulting from changes in ocean gateways may be poorly represented. Therefore, although these simulations are likely to capture large scale climate phenomena, clearly much could be learnt in future research from using higher resolution models."*

Nevertheless, these results bring some credit to the respective contributions of AIS development, CO2 decrease and Drake opening to the EOT and conclude to a minor role played by this latter. This conclusion is therefore interesting but remains not completely convincing due to many uncertainties that the authors describe.

*This has been further acknowledged:*

*"These results point towards some interesting conclusions about how the Earth system changed across the EOT, however this work remains a first step upon which further research should be built."*

The constraints on the development of the AIS when coupling climate and cryosphere during the EOT are also a promising way to capture the evolution of a whole system in which the AIS is active. Increasing model resolution, coupling GCM and ice sheet models, increasing the number of data through EOT, better accounting for annual temperature variability induced by astronomical forcing factors, all these features are important ways to continue this pioneering study.

*This has been included in the discussion of how the work could be improved in the future:*
*"as well as fully accounting for uncertainty in seasonal biases and orbital variations"*

Figures:

On fig1, for sites that benefited from several temperature reconstructions as Ross or Falklands Plateau, the spread is large which shows that a future work of intercalibration and synthesis like MARGO remains necessary. From fig. 1, it is clear that there is a tendency to lower temperatures for Oligocene, however, the values for both periods are compatible within the error bars. Indeed in fig.2, most of the values for the temperature differences are compatible with zero. However, there is a clearly negative value for UK37 , but it is not clear why the error bar for this value is so small.

We are unsure what U$_{K'37}$ value is being referred to here as the error bars are comparable to other proxies. In any case, the error bars will have been taken from published values.

figure 3: caution it is not North but South on the axis title.

This has been changed.

Despite the large uncertainty of the models, the models tend to underestimate the high latitude temperatures (fig. 3 a and b), and the effect is larger for Oligocene. Hence, the models predict in overall a larger cooling at high latitude than the data (fig 3c). The surprising behavior depicted in the text (larger cooling over mid latitude than for high latitudes) is indeed not reproduced by the model. However, the signal for such a latitudinal trend seems very weak and considering the model uncertainty and the error bars on the data, the model results seem to be compatible with each data point.

Discussion of the differences in the latitudinal gradient for the Eocene and Oligocene (Figure 3a and 3b) has been adjusted reflecting that the difference between the time slices is insignificant and dependent on a few extreme (and uncertain points):

*"In addition to the latitudinal gradient calculated for the full dataset for each time slice, an uncertainty range for the gradient was calculated by systematically omitting single points from the comparison. The absolute temperature profiles in the late Eocene and early Oligocene proxy datasets show colder temperatures at higher latitudes than mid-latitudes, as would be expected. The latitudinal gradient is comparable between the early Oligocene (0.54 °C °N-1; range 0.30 to 0.63 °C °N-1) and the late Eocene (0.49 °C °N-1; range 0.45 to 0.54 °C °N-1), with the Oligocene gradient more uncertain due to the greater variability in the proxy records."*

However, the EOT latitudinal gradient (Figure 3c) persist for all combinations of data (i.e. are not purely influenced by extreme values):

[revised manuscript text omitted]